# Connecting NTK and NNGP: A Unified Theoretical Framework for Neural Network Learning Dynamics in the Kernel Regime

## Abstract

Artificial neural networks (ANNs) have revolutionized machine learning in recent years, but a complete theoretical framework for their learning process is still lacking. Substantial theoretical advances have been achieved for infinitely wide networks. In this regime, two disparate theoretical frameworks have been used, in which the network's output is described using kernels: one framework is based on the Neural Tangent Kernel (NTK), which assumes linearized gradient descent dynamics, while the Neural Network Gaussian Process (NNGP) kernel assumes a Bayesian framework. However, the relation between these two frameworks and between their underlying sets of assumptions has remained elusive. This work unifies these two distinct theories using gradient descent learning dynamics with an additional small noise in an ensemble of randomly initialized infinitely wide deep networks. We derive an exact analytical expression for the network input-output function during and after learning and introduce a new time-dependent Neural Dynamical Kernel (NDK) from which both NTK and NNGP kernels can be derived. We identify two important learning phases characterized by different time scales: gradient-driven and diffusive learning. In the initial gradient-driven learning phase, the dynamics is dominated by deterministic gradient descent, and is adequately described by the NTK theory. This phase is followed by the slow diffusive learning stage, during which the network parameters sample the solution space, ultimately approaching the equilibrium posterior distribution corresponding to NNGP. Combined with numerical evaluations on synthetic and benchmark datasets, we provide novel insights into the different roles of initialization, regularization, and network depth, as well as phenomena such as early stopping and representational drift. This work closes the gap between the NTK and NNGP theories, providing a comprehensive framework for understanding the learning process of deep neural networks in the infinite width limit.

## 1 Introduction

Despite the empirical success of artificial neural networks (ANNs), theoretical understanding of their underlying learning process is still limited. One promising theoretical approach focuses on deep wide networks, in which the number of parameters in each layer goes to infinity while the number of training examples remains finite (Jacot et al. (2018); Lee et al. (2018; 2019); Novak et al. (2018; 2019); Matthews et al. (2018); Yang (2019); Sohl-Dickstein et al. (2020)). In this regime, the neural network (NN) is highly over-parameterized, and there is a degenerate space of solutions achieving zero training error. Investigating the properties of the solution space offers an opportunity for understanding learning in over-parametrized NNs (Chizat & Bach (2020); Jin & Montúfar (2020); Min et al. (2021)). The two well-studied theoretical frameworks in the infinite width limit focus on two different scenarios for exploring the solution space during learning. One considers randomly initialized NNs trained with gradient descent dynamics, and the learned NN parameters are largely dependent on their value at initialization. In this case, the infinitely wide NN's input-output relation is captured by the neural tangent kernel (NTK) (Jacot et al. (2018); Lee et al. (2019)). The other scenario considers Bayesian neural networks (BNNs) with an i.i.d. Gaussian prior over their parameters, and a learning-induced posterior distribution. In this case, the statistics of the NN's input-output relation in the infinite width limit are given by the neural network Gaussian

process (NNGP) kernel (Cho & Saul (2009); Lee et al. (2018)). These two scenarios make different assumptions regarding the learning process and regularization. Furthermore, for some datasets the generalization performance of the two kernels differs significantly (Lee et al. (2020)). It is therefore important to generate a unified framework with a single set of priors and regularizations describing a dynamical process that captures both cases. Such a theory may also provide insight into salient dynamical phenomena such as early stopping (Li et al. (2020); Advani et al. (2020); Ji et al. (2021)). From a neuroscience perspective, a better understanding of the exploratory process leading to Bayesian equilibrium may shed light on the empirical and hotly debated phenomenon of representational drift (Rokni et al. (2007); Rule et al. (2019); Deitch et al. (2021); Marks & Goard (2021); Schoonover et al. (2021)). To this end, we construct a new analytical theory of the learning dynamics in infinitely wide ANNs. Our main contributions are:

1. We derive an analytical expression for the time evolution of the mean input-output relation (i.e. the mean predictor) of infinitely wide networks under Langevin dynamics in the form of an integral equation, and demonstrate its remarkable agreement with computer simulations.

2. A new two-time kernel, the Neural Dynamical Kernel (NDK), naturally emerges from our theory and we derive explicit relations between the NDK and both the NTK and the NNGP kernels.

3. Our theory reveals two important learning phases characterized by different time scales: gradient-driven and diffusive learning. In the initial gradient-driven learning phase, the dynamics is primarily governed by deterministic gradient descent, and can be described by the NTK theory. This phase is followed by a slow diffusive stage, during which the network parameters sample the solution space, ultimately approaching the equilibrium posterior distribution corresponding to NNGP.

4. We apply our theory to both synthetic and benchmark datasets and present several predictions. Firstly, the generalization error may exhibit diverse behaviors during the diffusive learning phase depending on network depth and the ratio between initialization and regularization strengths. Our theory provides insights into the roles of these hyper-parameters in early stopping. Secondly, through analysis of the temporal correlation between network weights during diffusive learning, we show that despite the random diffusion of hidden layer weights, the training error remains stable at a very low value due to a continuous realignment of readout weights and network hidden layer weights. Conversely, a time delay in this alignment degrades the network performance due to decorrelation in the representation, ultimately leading to degraded performance. We derive conditions under which the performance upon completely decorrelated readout and hidden weights remain well above chance. This provides insight into the representational drift and its consequences observed in biological neuronal circuits.

5. Relation to previous work: Previous work considered a single-time NTK kernel Jacot et al. (2018). This implicit time dependence arises through the weight dependence of an uaveraged weight-dependent NTK. Our (two-) time dependence arises in an appropriate weight-averaged kernel, and therefore exhibits explicit time dependence. Other perspective on the two phases has been proposed by previous works, Shwartz-Ziv & Tishby (2017) establishes an information theory framework to describe the two phases Li et al. (2021); Blanc et al. (2020) analyze the two phases in SGD, where the diffusive phase is driven by the different types of noise (label noise or isotropic noise). Our work complements these previous findings by providing theoretical analysis of the two phases under Langevin dynamics, reaffirming the connections between the diffusive learning stage and representational drift in neuroscience as established in previous works Aitken et al. (2022); Pashakhanloo & Koulakov (2023).

## 2 THEORETICAL RESULTS

In this section, we present our dynamic theory for infinitely wide deep networks under Langevin dynamics. We define a new time-dependent kernel, the Neural Dynamical Kernel (NDK), and derive an exact analytical integral equation for the mean predictor of the network.

### 2.1 NOTATION AND SETUP FOR ARCHITECTURE AND TRAINING DYNAMICS

We consider a fully connected DNN with $L$ hidden layers with a vector input $\mathbf{x} \in \mathbb{R}^{N_0}$ and a single output $\mathrm{f}(\mathbf{x}, t)$ (i.e., the predictor), with the following time-dependent input-output function:

$$\mathrm{f}(\mathbf{x}, t) = \frac{1}{\sqrt{N_L}} \mathbf{a}(t) \cdot \mathbf{x}^L(\mathbf{x}, t), \quad \mathbf{a}(t) \in \mathbb{R}^{N_L} \tag{1}$$

$$\mathbf{x}^l(\mathbf{x}, t) = \phi\left(N_{l-1}^{-1/2} \mathbf{W}^l(t) \cdot \mathbf{x}^{l-1}(\mathbf{x}, t)\right), \quad \mathbf{x}^l(\mathbf{x}, t) \in \mathbb{R}^{N_l} \quad , l = 1, \cdots, L \tag{2}$$

Here $N_l$ denotes the number of nodes in hidden layer $l$, and $N_0$ is the input dimension. The set of network weights at a training time $t$ is denoted collectively as $\boldsymbol{\theta}\left(t\right) = \left\{\mathbf{W}^1\left(t\right)\cdots\mathbf{W}^L\left(t\right), \mathbf{a}\left(t\right)\right\}$, where $\mathbf{a}\left(t\right) \in \mathbb{R}^{N_L}$ denotes the linear readout weights and $\mathbf{W}\left(t\right)^l \in \mathbb{R}^{N_l \times N_{l-1}}$ the hidden layer weights between layer $l-1$ and $l$. $\phi\left(N_{l-1}^{-1/2}\mathbf{W}^l\left(t\right) \cdot \mathbf{x}^{l-1}\left(\mathbf{x}, t\right)\right)$ is an element-wise nonlinear function of the weighted sum of its input vector, and $\mathbf{x}^{l=0} \equiv \mathbf{x}$ is the input to the first layer. The training data is a set of $P$ labeled examples $\mathcal{D} : \left\{\mathbf{x}^\mu, y^\mu\right\}_{\mu=1,\cdots,P}$ where there are P training input vectors $\mathbf{x}^\mu \in \mathbb{R}^{N_0}$, and $\boldsymbol{y} \in \mathbb{R}^P$ is a vector of the target labels of the training examples. We denote $\mathbf{f}_{\text{train}}\left(t\right) \in \mathbb{R}^P$, a vector containing the predictor on the $P$ training vectors. We consider the supervised learning cost function:

$$E\left(\boldsymbol{\theta}\left(t\right)|\mathcal{D}\right) = \frac{1}{2}\left|\mathbf{f}_{\text{train}}\left(t\right) - \boldsymbol{y}\right|^2 + \frac{T}{2\sigma^2}\left|\boldsymbol{\theta}\left(t\right)\right|^2 \tag{3}$$

The first term is the squared error empirical loss (SE loss), and the second term is a regularization term that favors weights with small $L_2$ norm, where $\left|\boldsymbol{\theta}\left(t\right)\right|^2$ is the sum of the squares of all weights. It is convenient to introduce the temperature parameter (see definition below) $T$ as controlling the relative strength of the regularization, and $\sigma^2$ is the variance of the equilibrium distribution of the Gaussian prior. We consider noisy gradient descent learning dynamics given by continuous-time Langevin dynamics, where the weights of the system start from an i.i.d. Gaussian initial condition with zero mean and variance $\sigma_0^2$, and evolve under gradient descent dynamics with respect to the cost function above with an additive white noise $\boldsymbol{\xi}\left(t\right)$:

$$\frac{d}{dt}\boldsymbol{\theta}\left(t\right) = -\nabla_{\boldsymbol{\theta}}E\left(\boldsymbol{\theta}\left(t\right)\right) + \boldsymbol{\xi}\left(t\right) \tag{4}$$

where $\boldsymbol{\xi}\left(t\right)$ has a white noise statistics $\mathbb{E}\left[\boldsymbol{\xi}\left(t\right)\boldsymbol{\xi}\left(t'\right)^\top\right] = 2T\boldsymbol{I}\delta\left(t-t'\right)$, $\mathbb{E}\left[\boldsymbol{\xi}\left(t\right)\right] = 0$, and $T$ is the temperature controlling the level of noise in the system. As we show below, the additional white noise compared to deterministic gradient descent allows for continued exploration of the solution space after reaching a small training error, and enables the connection from NTK to NNGP theory.

## 2.2 INFINITE WIDTH LIMIT

We are interested in the predictor statistics (in particular the mean predictor) induced by the Langevin dynamics, which can be evaluated analytically in the infinite width where the hidden layer widths are taken to infinity, while the number of training examples $P$ remains finite. For simplicity, we consider all the $N_l$ to be the same for $l = 1, \cdots, L$ and equal to $N$, $N \to \infty$.

SI Sec. A presents a derviation of a path integral formulation of the above Langevin dynamics using a Markov proximal learning framework. Evaluating statistical quantities using these integrals is in general intractable. However, in the infinite width limit, they become tractable, as proven in SI Sec. B.1-B.4. Specifically, the moments of the predictor can be derived from a moment generating function (MGF) $\mathcal{M}\left[\ell(t)\right]$, written in the form of a path integral over two auxiliary time-dependent vectors, $\mathbf{u}\left(t\right) \in \mathbb{R}^P$ and $\mathbf{v}\left(t\right) \in \mathbb{R}^P$. Additionally, $\tilde{\mathbf{u}}\left(t\right) = \left[\mathbf{u}(t), i\ell(t)\right] \in \mathbb{R}^{P+1}$, where $\ell(t)$ denotes the field coupled to the predictor $f(\mathbf{x}, t)$ on an arbitrary test input vector $\mathbf{x}$. Therefore, the moments of the predictor can be derived by evaluating the derivative of $\mathcal{M}[\ell(t)]$ at $\ell(t) = 0$. In SI Sec. B.4 it is shown that the MGF has the following form,

$$\mathcal{M}\left[\ell(t)\right] = \int D\mathbf{u}(t)\int D\mathbf{v}(t)\exp\left(-S\left[\mathbf{v}(t), \tilde{\mathbf{u}}(t)\right]\right) \tag{5}$$

$$S\left[\mathbf{v}\left(t\right), \tilde{\mathbf{u}}\left(t\right)\right] = \frac{1}{2}\int_0^\infty dt \int_0^\infty dt' m\left(t, t'\right)\tilde{\mathbf{u}}^\top\left(t\right)\tilde{\mathbf{K}}^L\left(t, t'\right)\tilde{\mathbf{u}}\left(t'\right) \tag{6}$$

$$+ \int_0^\infty dt \int_0^t dt'\tilde{\mathbf{u}}\left(t\right)^\top \tilde{\mathbf{K}}^{d,L}\left(t, t'\right)\mathbf{v}\left(t'\right) + \int_0^\infty dt\mathbf{u}(t)^\top\left(\mathbf{v}\left(t\right) - i\boldsymbol{y}\right)$$

where $\int D\mathbf{u}(t)$ means integration over all trajectories of $\mathbf{u}$ and similarly for $\mathbf{v}$. The two-time kernel matrices $\tilde{\mathbf{K}}^L(t, t') \in \mathbb{R}^{(P+1)\times(P+1)}$ and $\tilde{\mathbf{K}}^{d,L}(t, t') \in \mathbb{R}^{(P+1)\times P}$ in Eq. 6 are defined by applying the kernel functions $K^L(t, t', \mathbf{x}, \mathbf{x}')$ and $K^{d,L}(t, t', \mathbf{x}, \mathbf{x}')$ on $P$ training data $\mathbf{x}_\mu, 1 \le \mu \le P$ and a single test point, $\mathbf{x}_{P+1} = \mathbf{x}$. Specifically, $\tilde{\mathbf{K}}_{\mu,\nu}^L(t, t') = \mathbf{K}^L(\mathbf{x}_\mu, \mathbf{x}_\nu, t, t'), 1 \le \mu, \nu \le P+1$ and

$\tilde{\mathbf{K}}_{\mu,\nu}^{d,L} = \tilde{\mathbf{K}}^{d,L}(\mathbf{x}_\mu, \mathbf{x}_\nu, t, t'), 1 \le \mu \le P + 1, 1 \le \nu \le P$. $m(t,t')$ is a two-time dependent scalar function.

We first provide expressions of these kernel functions as well as the scalar two-time function $m(t,t')$ in Sec. 2.3; we then present an expression for the mean predictor in terms of these kernels in Sec. 2.4. The full derivation of these equations are given in Sec. B.1-B.4.

## 2.3 THE NEURAL DYNAMICAL KERNEL (NDK)

The kernel function $K^{d,L}(t,t',\mathbf{x},\mathbf{x}')$ in Eq. 6 is a new kernel function in our theory, denoted as the Neural Dynamical Kernel (NDK), which can be viewed as a time-dependent generalization of the NTK, and can be expressed in terms of derivatives of the predictor w.r.t. the time-dependent network parameters (SI Sec. C.4):

$$K^{d,L}(t,t',\mathbf{x},\mathbf{x}') = e^{-T\sigma^{-2}|t-t'|}\mathbb{E}_{\boldsymbol{\theta}\sim S_0}\left[\nabla_{\boldsymbol{\theta}(t)}\mathbf{f}(\mathbf{x},t) \cdot \nabla_{\boldsymbol{\theta}(t')}\mathbf{f}(\mathbf{x}',t')\right] \tag{7}$$

where $S_0$ denotes a Gaussian probability measure of the weights

$$\mathbb{E}_{\boldsymbol{\theta}\sim S_0}\left[\boldsymbol{\theta}(t)\boldsymbol{\theta}(t')^\top\right] = m(t,t')\boldsymbol{I}, \mathbb{E}_{\boldsymbol{\theta}\sim S_0}\left[\boldsymbol{\theta}(t)\right] = 0 \tag{8}$$

$$m(t,t') = \sigma^2 e^{-T\sigma^{-2}|t-t'|} + \left(\sigma_0^2 - \sigma^2\right)e^{-T\sigma^{-2}(t+t')} \tag{9}$$

Here and in Eq. 7 $T$ is the level of noise in the Langevin dynamics, $\sigma^2$ and $\sigma_0^2$ are the variances of the $L_2$ regularizer and initial weights distribution, respectively. As expected, $m(0,0) = \sigma_0^2$ is the variance of the weights at initialization. At long times, the last (transient) term in Eq. 9 vanishes and the first term dominates, such that $m(t,t')$ and $K^{d,L}(t,t',\mathbf{x},\mathbf{x}')$ become functions of time difference $|t - t'|$. From Eqs. 7, 9 follow that at initialization

$$K^{d,L}(0,0,\mathbf{x},\mathbf{x}') = \mathbb{E}_{\boldsymbol{\theta}_0\sim\mathcal{N}(0,\boldsymbol{I}\sigma_0^2)}\left[\nabla_{\boldsymbol{\theta}_0}\mathbf{f}(\mathbf{x},0) \cdot \nabla_{\boldsymbol{\theta}_0}\mathbf{f}(\mathbf{x}',0)\right] = K_{NTK}^L(\mathbf{x},\mathbf{x}') \tag{10}$$

The NDK equals the NTK as the average is only over the i.i.d. Gaussian initialization. Furthermore, as we will see in Sec. 3.2, the NNGP kernel can also be obtained from the NDK. The other kernel function $K^L(t,t',\mathbf{x},\mathbf{x}')$ that appears in Eq. 6 is a two-time extension of the NNGP kernel function

$$K^L(t,t',\mathbf{x},\mathbf{x}') = \mathbb{E}_{\boldsymbol{\theta}\sim S_0}\left[N^{-1}\mathbf{x}^L(\mathbf{x},t) \cdot \mathbf{x}^L(\mathbf{x}',t')\right] \tag{11}$$

The NDK defined in Eq. 7 can be computed recursively, in terms of the two-time NNGP kernel $K^L(t,t',\mathbf{x},\mathbf{x}')$ and the derivative kernel $\dot{K}^L(t,t',\mathbf{x},\mathbf{x}')$ (see SI 2.3 for a detailed proof of the equivalence), given by

$$K^{d,L}(t,t',\mathbf{x},\mathbf{x}') = m(t,t')\dot{K}^L(t,t',\mathbf{x},\mathbf{x}')K^{d,L-1}(t,t',\mathbf{x},\mathbf{x}') + e^{-T\sigma^{-2}|t-t'|}K^L(t,t',\mathbf{x},\mathbf{x}')$$

$$K^{d,L=0}(t,t',\mathbf{x},\mathbf{x}') = e^{-T\sigma^{-2}|t-t'|}\left(N_0^{-1}\mathbf{x}\cdot\mathbf{x}'\right) \tag{12}$$

The derivative kernel, $\dot{K}^L(t,t',\mathbf{x},\mathbf{x}')$ is the kernel evaluated w.r.t. the derivative of the activation functions

$$\dot{K}^L(t,t',\mathbf{x},\mathbf{x}') \equiv \mathbb{E}_{\boldsymbol{\theta}\sim S_0}\left[N^{-1}\dot{\mathbf{x}}^L(\mathbf{x},t) \cdot \dot{\mathbf{x}}^L(\mathbf{x}',t')\right] \tag{13}$$

where $\dot{\mathbf{x}}^L(\mathbf{x},t) = \phi'\left(N_{L-1}^{-\frac{1}{2}}\mathbf{W}_t^L \cdot \mathbf{x}^{L-1}(\mathbf{x},t)\right)$, namely $d\phi(z)/dz$ evaluated at the preactivtion of $\mathbf{x}^{L-1}$. All the kernel functions above including the two-time NNGP kernel, the derivative kernel, and the NDK, have closed-form expressions for specific activation functions such as linear, ReLU and error function (see SI Sec. C.1-C.3, Cho & Saul (2009); Williams (1996)).

## 2.4 EQUATIONS FOR THE MEAN PREDICTOR

Here we provide the expressions of the mean predictor averaged over the distribution of learning trajectories, under the Langevin dynamics (Eq. 4), using the NDK introduced above. The mean predictor can be derived by evaluating the derivative of the MGF (Eq. 37, see details in SI Sec. B.3) The mean predictor on the training inputs obeys the following integral equation

$$\mathbb{E}\left[\mathbf{f}_{\text{train}}(t)\right] = \int_0^t dt' \mathbf{K}^{d,L}(t,t')\left(\boldsymbol{y} - \mathbb{E}\left[\mathbf{f}_{\text{train}}(t')\right]\right) \tag{14}$$

and the mean predictor on any test point $\mathbf{x}$ is given by an integral over the training predictor with the NDK of the test data $\mathbf{x}$

$$\mathbb{E}\left[\mathrm{f}\left(\mathbf{x}, t\right)\right] = \int_0^t dt' \mathbf{k}^{d,L}\left(t, t'\right)^\top \left(\boldsymbol{y} - \mathbb{E}\left[\mathbf{f}_{\mathrm{train}}\left(t'\right)\right]\right) \tag{15}$$

Here we have introduced separate notations for the kernel function applied on training data and testing data, $\mathbf{K}^{d,L}(t, t') \in \mathbb{R}^{P \times P}$ and $\mathbf{k}^{d,L}(t, t') \in \mathbb{R}^P$, defined as $\mathbf{K}^{d,L}_{\mu,\nu}(t, t') \equiv \tilde{\mathbf{K}}^{d,L}_{\mu,\nu}(t, t')$ and $\mathbf{k}^{d,L}_\mu(t, t') \equiv \tilde{\mathbf{K}}^{d,L}_{P+1,\mu}(t, t')$, respectively. Throughout the paper we will use similar notations for these kernel matrices and vectors (i.e., $\mathbf{K} \in \mathbb{R}^{P \times P}$ for kernel functions applied on training data, and $\mathbf{k} \in \mathbb{R}^P$ for kernel functions applied on test and training data).

## 3 CORRESPONDENCE TO NTK AND NNGP

In this section, we show that our theory recovers known results of the NTK (Jacot et al. (2018)) and NNGP theories (Lee et al. (2018)) in the short- and long-time limit, respectively. We stress that the separation of time scales occurs in the limit of small but nonzero noise controlled by $T$, which is also the relevant limit of a realistic machine-learning scenario.

### 3.1 GRADIENT-DRIVEN PHASE CORRESPONDS TO NTK DYNAMICS

The time dependence of the NDK (Eq. 12) is in time scales of $T \cdot t$ (Eqs. 9, 12), and thus at low $T$ and $t \sim \mathcal{O}(1)$ we can substitute $K^{d,L}(t, t', \mathbf{x}, \mathbf{x}') = K^{d,L}(0, 0, \mathbf{x}, \mathbf{x}')$. In Sec. 2.3, we obtain an exact equivalence between the NDK at time zero and the NTK. In this regime, $\mathbf{K}^{d,L}(t, t')$ and $\mathbf{k}^{d,L}(t, t')$ in Eq. 15 and Eq. 14 become time-independent. By taking the time derivative on both sides, the integral equations Eq. 15 and Eq. 14 can be transformed into a linear ODE, and solved analytically, leading to the well-known mean predictor in the NTK theory:

$$\mathbb{E}\left[\mathrm{f}\left(\mathbf{x}, t\right)\right] \approx \mathbf{k}^{L}_{NTK}{}^\top \left[\mathbf{K}^L_{NTK}\right]^{-1} \left(I - \exp\left(-\mathbf{K}^L_{NTK} t\right)\right) \boldsymbol{y}, t \sim \mathcal{O}(1) \tag{16}$$

where we define $\mathbf{k}^L_{NTK} \in \mathbb{R}^P$ and $\mathbf{K}^L_{NTK} \in \mathbb{R}^{P \times P}$ as the NTK applied on test and training data, respectively, similar to Sec. 2.3. We see that the NTK theory describes the dynamics of the system when the time is short compared to the level of noise, such that the dynamics is approximately deterministic. Taking the large $t$ limit of the NTK dynamics (Eq. 16) results in the "NTK equilibrium", where $\lim_{t \to \infty} \mathbb{E}\left[\mathrm{f}\left(\mathbf{x}, t\right)\right] = \mathbf{k}^L_{NTK}{}^\top \left[\mathbf{K}^L_{NTK}\right]^{-1} \boldsymbol{y}$. This short-time equilibrium marks the crossover between the gradient-driven phase and the diffusive learning phase. After the NTK equilibrium point, the gradient of the loss is $\mathcal{O}(T)$, and thus the two parts of the cost function in Eq. 3 (the SE loss and the regularization) are on equal footing, and give rise to the diffusive dynamics in time scales of $t \sim \mathcal{O}(T^{-1})$.

### 3.2 LONG-TIME EQUILIBRIUM CORRESPONDS TO NNGP

Now we investigate the behavior at long time scales defined by $t, t' \gg T^{-1}$. In this regime, $K^{d,L}(t, t', \mathbf{x}, \mathbf{x}') = K^{d,L}(t - t', \mathbf{x}, \mathbf{x}')$ is a function of the time difference, and the transient dependence on the initialization parameter $\sigma_0$ vanishes. Furthermore, in this regime the limit of the integral of the NDK (Eq. 7, Eq. 12) satisfies the following identity (see SI Sec. C.4 for detailed proof):

$$\lim_{t \to \infty} \left(\int_0^t K^{d,L}\left(t - t', \mathbf{x}, \mathbf{x}'\right) dt'\right) = \sigma^2 T^{-1} K^L_{GP}\left(\mathbf{x}, \mathbf{x}'\right) \tag{17}$$

where $K^L_{GP}\left(\mathbf{x}, \mathbf{x}'\right) = \mathbb{E}_{\mathbf{W} \sim \mathcal{N}(0, \sigma^2 I)}\left[N^{-1}\mathbf{x}^L\left(\mathbf{x}\right) \cdot \mathbf{x}^L\left(\mathbf{x}'\right)\right]$ is the well-known NNGP kernel. As a result, the mean predictor on arbitrary input $\mathbf{x}$ (Eq. 15) correspondingly converges to an equilibrium (see SI C.4),

$$\lim_{t \to \infty} \mathbb{E}\left[\mathrm{f}\left(\mathbf{x}, t\right)\right] = \mathbf{k}^L_{GP}{}^\top \left(\boldsymbol{I} T \sigma^{-2} + \mathbf{K}^L_{GP}\right)^{-1} \boldsymbol{y} \tag{18}$$

where $\mathbf{k}^L_{GP} \in \mathbb{R}^P$ is the NNGP kernel function applied to $\mathbf{x}$. This is the known equilibrium NNGP result (Lee et al. (2018)). We emphasize that this result is true for any temperature, while the NTK solution in Sec. 3.1 is relevant at low $T$ only. Our theory thus establishes the connection between the NTK and the NNGP theories.

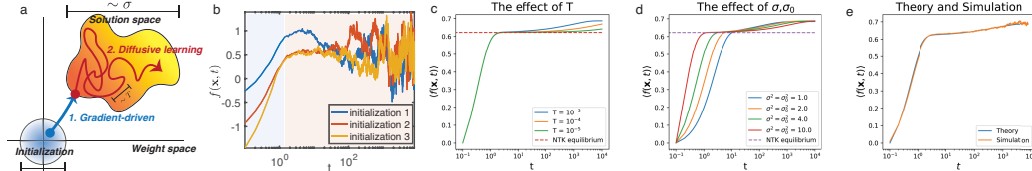

Figure 1: Two phases of the dynamics. Example using a synthetic dataset where the training inputs are orthogonal to each other with random binary labels $y^\mu \in \{\pm 1\}$. Each test point has partial overlap with one input point and is orthogonal to all the others. The desired test label is the same as the label on the training input with which it has nonzero overlap. (a) Schematics of the dynamics, the weights are initialized with width $\sigma_0$. The gradient-driven dynamics bring the weights to the solution space with a small training error, and the diffusive learning dynamics explores the solution space with a time scale $T^{-1}$. (b) Three example trajectories of $f(\mathbf{x}, t)$, the dynamics are initially deterministic, and fluctuate significantly when $t$ is large. (c-d) The network mean predictor on a test point with the desired label $+1$ (see details in SI Sec.E). (c) $T$ does not affect the initial gradient-driven phase, but decreasing $T$ slows the dynamics of the diffusive learning phase. (d) Increasing $\sigma^2$ and $\sigma_0^2$ simultaneously (keeping $\sigma^2 = \sigma_0^2$) affects the time scales of the two phases differently. The time scale of the gradient-driven phase decreases as $\sigma_0^2$ increases and vice versa in the diffusive dynamics. (e) The mean predictor under Langevin simulations of neural networks for the synthetic dataset agrees well with the theory prediction.

## 4 DYNAMICS AT LOW $T$

In this section, we study the equations for the mean predictor dynamics (Eqs. 14, 15) in the important limit of low $T$. As we show below and illustrate in Fig.1(a,b), the network dynamics exhibits two distinct regimes. First, the network weights are initialized with width $\sigma_0$, and converge to weights with almost zero training error (error of $\mathcal{O}(T)$) approximately deterministically. Subsequently, the network executes slow and noise-driven explorations (on a time scale of $\mathcal{O}(T^{-1})$) of the solution space, regularized by a Gaussian prior with width $\sigma$. We investigate how the different parameters such as initialization, regularization and the level of noise affect the learning behavior by evaluating numerically Eqs. 14, 15.

### 4.1 TIME SCALES OF THE DYNAMICS

In this section, we further examine how the time scales of the dynamics in the two phases are affected by the different hyper-parameters. We focus on the level of stochasticity $T$, the initialization ($\sigma_0^2$), and regularization ($\sigma^2$). As can be seen in Eqs.9, 12, the dynamics depend on $t$ through exponents $\exp\left(-T\sigma^{-2}t\right)$ and a scalar factor that depends on $\sigma_0^2/\sigma^2$. To determine the time scales of the dynamics, we fix the scalar factor $\sigma_0^2/\sigma^2$ as a constant as we vary $\sigma_0^2, \sigma^2$ and $T$ respectively. We consider $\sigma_0^2, \sigma^2 \sim \mathcal{O}(1)$.

First, we evaluate how the dynamics depends on the level of stochasticity determined by a small but nonzero $T$. As we see in Fig.1 (c), while the initial learning phase is not affected by $T$ since the dynamics is mainly driven by deterministic gradient descent, the diffusive phase is slower for smaller $T$ since it is driven by noise. We then investigate how the dynamics depends on $\sigma^2$ and $\sigma_0^2$ while fixing the ratio between them. Fig.1 (d) shows that as we increase $\sigma^2$ and $\sigma_0^2$ simultaneously, the gradient dynamics becomes faster since the initialization weights determined by $\sigma_0^2$ are closer to the typical solution space (with the $L_2$ regularization), while the dynamics of the diffusive phase becomes slower since the regularization determined by $\sigma^2$ imposes less constraint on the solution space, hence exploration time increases.

### 4.2 DIFFUSIVE LEARNING DYNAMICS EXHIBIT DIVERSE BEHAVIORS

In this section, we focus on the diffusive phase, where $t \sim \mathcal{O}(1/T)$. Unlike the simple exponential relaxation of the gradient-driven stage, in the diffusive phase, the predictor dynamics exhibits complex behavior dependent on depth, regularization, initialization and the data. We systematically explore these behaviors by solving the integral equations (Eqs.14, 15) numerically for benchmark datasets as well as a simplified synthetic task (see details of the tasks in Fig.1, 2 captions and SI Sec.E). We verify the theoretical predictions with simulations of the gradient-based Langevin dynamics of finite width neural networks with sufficiently small discretisation time step Alfonsi et al. (2015), as shown in Fig.1(e) and SI Sec.F. Even though in the diffusive phase, the dominant dy-

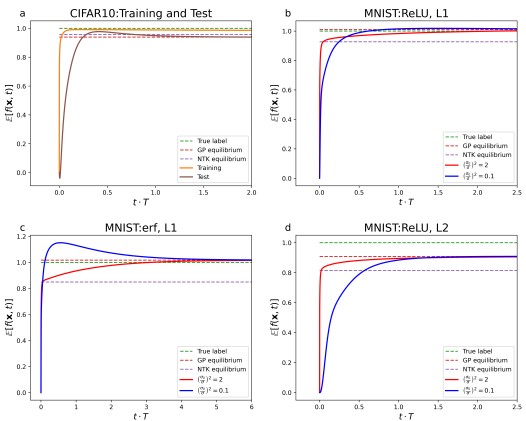

Figure 2: Dynamics of the mean predictor on a given test point in benchmark datasets. All test points shown have a target label $+1$. (a) Result on CIFAR10 dataset (Krizhevsky et al. (2014)) with binary classification of cats vs dogs, for $\sigma_0^2/\sigma^2 = 2$. We see a fast convergence of the mean predictor on the training point while the test point exhibits a diffusive learning phase on time scales $t \sim \mathcal{O}(1/T)$. (b-d) Results on MNIST dataset (Deng (2012)) with binary classification of 0 vs 1 digits, for $L = 1, 2$. In $L = 2$ the effect of $\sigma_0^2/\sigma^2$ is larger. (d) Results on MNIST dataset in a network with an error function (erf) nonlinearity with a single hidden layer. The effect $\sigma_0^2/\sigma^2$ is significantly larger than in (b,c).

namics is driven by noise and the regularization, the learning signal (both on the readout weights and the hidden layers) from the gradient of the loss is what restricts the exploration to the subspace of low ($\mathcal{O}(T)$) training error, and without it the performance will deteriorate back to chance.

**The role of initialization and regularization and early stopping phenomena:** We investigate how the diffusive dynamics is affected by the $\sigma_0^2$ for fixed values of $\sigma^2$ and $T$ (thus fixing the time scale of the diffusive learning phase). As expected, the training predictor converges fast to the target output and exhibits little deviation afterward (see Fig. 2 (a)). In the previous section, we kept the ratio $\sigma_0^2/\sigma^2$ fixed, resulting in the same qualitative behavior with different time scales. In Fig. 2 (b-d) , we show that changing the ratio $\sigma_0^2/\sigma^2$ results in qualitatively different behaviors of the trajectory, shown across network depth and nonlinearities. Interestingly, in Fig. 2 (c), when $\sigma_0^2/\sigma^2$ is small, the predictor dynamics is non-monotonic, overshooting above its equilibrium value. The optimal early stopping point, defined as the time the network reaches the minimal generalization error in the entire learning trajectory from $t = 0$ to $t \to \infty$, occurs in the diffusive learning phase. In this case, the performance in the diffusive phase is better than both equilibria. We study the effect of $\sigma_0^2/\sigma^2$ on the early stopping point systematically in the synthetic dataset in Fig. 3.

**The role of depth:** The effect of different $\sigma_0^2/\sigma^2$ ratios on the dynamics increases with depth, resulting in distinctively different behavior for different ratios (Fig. 2 (b,d)). Depth also changes the NTK and NNGP equilibrium, typically in favor of the NNGP solution as the network grows deeper (see SI Sec. D.1). Furthermore, as shown in Fig. 3, depth also has an effect on the occurrence of the optimal early stopping time. In the synthetic dataset, the early stopping time occurs earlier in shallower networks for small $\sigma_0^2/\sigma^2$, and does not occur when $L > 3$.

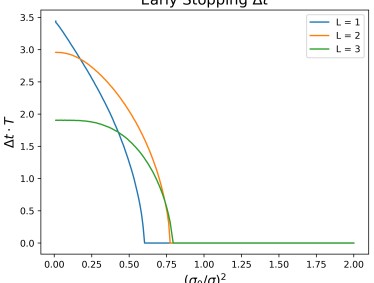

Figure 3: The time difference between the optimal stopping time and the long-time equilibrium time scaled by $T$ (denoted by $\Delta t \cdot T$), for the synthetic orthogonal dataset in networks with hidden layers $L = 1, 2, 3$. We see that for small $\sigma_0^2/\sigma^2$ the optimal stopping time occurs during the diffusive learning phase, while for large $\sigma_0^2/\sigma^2$ the optimal stopping time is only at the long-time equilibrium, which corresponds to the NNGP. Interestingly, in this dataset for $L > 3$ there is no early stopping point.

**The role of nonlinearity:** We compare the behaviors of networks with ReLU and error function, with both having closed-form expressions for their NDK (see SI C.1-C.3). As shown in Fig. 2 (c) with error function nonlinearity, the difference between NTK and NNGP is larger and the effect of $\sigma_0^2/\sigma^2$ on the network dynamics is more significant.

## 5 REPRESENTATIONAL DRIFT

We now explore the implications of the diffusive learning dynamics on the phenomenon of representational drift. Representational drift refers to neuroscience observations of neuronal activity patterns

accumulating random changes over time without noticeable consequences on the relevant animal behavior. These observations raise fundamental questions about the causal relation between neuronal representations and the underlying computation. Some of these observations were in the context of learned behaviors and learning-induced changes in neuronal activity. One suggestion has been that changes in the readout of the circuit compensate for the representational drift, leaving intact its input-output relation (Rule et al. (2020); Rule & O'Leary (2022)). We provide a general theoretical framework for studying such dynamics. In our model, the stability of the (low) training error during the diffusion phase, is due to the continuous realignment of readout weights $\mathbf{a}(t)$ to changes in the network hidden layer weights $\mathbf{W}(t)$ as they drift simultaneously exploring the space of solutions.

The above alignment scenario requires an ongoing learning signal acting on the weights. To highlight the importance of this signal, we consider an alternative scenario where the readout weights are frozen at some time (denoted as $t_0$) after achieving a low training error while the weights of the hidden layers $\mathbf{W}(t)$ continue to drift randomly without an external learning signal. We will denote the output of the network in this scenario as $\mathrm{f_{drift}}(\mathbf{x}, t, t_0)$. Our formalism allows for computation of the mean of $\mathrm{f_{drift}}(\mathbf{x}, t, t_0)$ (see SI Sec. D for details). We present here the results for large $t_0$, i.e., after the learning has finished.

$$\mathbb{E}\left[\mathrm{f_{drift}}(t - t_0)\right] = \left(\mathbf{k}^L(t - t_0)\right)^\top \left(\boldsymbol{I} T \sigma^{-2} + \mathbf{K}_{GP}^L\right)^{-1} \boldsymbol{y} \tag{19}$$

The kernel $\mathbf{k}^L(t - t_0)$ represents the overlap between the representations of the training inputs at time $t_0$ and that of a test point at time $t$. When $t - t_0$ is large, the two representations completely decorrelate and the predictor is determined by a new kernel $\tilde{K}_{mean}^L(\mathbf{x}, \mathbf{x}')$ defined as

$$K_{mean}^L(\mathbf{x}, \mathbf{x}') = N^{-1}\mathbb{E}_{\theta \sim \mathcal{N}(0, I\sigma^2)}\left[\mathbf{x}^L(\mathbf{x})\right] \cdot \mathbb{E}_{\theta \sim \mathcal{N}(0, I\sigma^2)}\left[\mathbf{x}^L(\mathbf{x}')\right] \tag{20}$$

which is a modified version of the NNGP kernel where the Gaussian averages are performed separately for each data point.

$$\lim_{t - t_0 \to \infty} \mathbb{E}\left[\mathrm{f_{drift}}(\mathbf{x}, t - t_0)\right] = \mathbf{k}_{mean}^L{}^\top \left(\boldsymbol{I} T \sigma^{-2} + \mathbf{K}_{GP}^L\right)^{-1} \boldsymbol{y} \tag{21}$$

where $\mathbf{k}_{mean}^L$ is defined as applying the mean kernel function to the test data. For some nonlinearities (e.g., linear and error function activation) $K_{mean}^L(\mathbf{x}, \mathbf{x}')$ is zero. This however, is not the case for other nonlinearities (e.g., ReLU). In these cases, its value depends on the input vectors' norms $\|\mathbf{x}\|, \|\mathbf{x}'\|$. Thus, if the distribution of the norms is informative of the given task, the predictor can still be useful despite the drift process. In this case, we can say that the norms are drift-invariant information. In other cases, the norms may not be relevant to the task, in which case the decorrelated output will yield a chance-level performance. We present examples for both scenarios in Fig. 4. We consider two MNIST binary classification tasks, after reaching the long-time equilibrium. For each one, we show the evolution of the histograms of the predictor on the training examples at times $t$, after freezing readout weights at an earlier time $t_0$. We train a linear classifier on top of the training predictors to evaluate the classification accuracy (see SI Sec. D for details). In the case of the classification task of the digit pair 4,9, the two histograms eventually overlap each other, resulting in a long-time chance level accuracy and a complete loss of the learned information. In contrast, in the classification of the digit pair 0,1 (Fig. 4(f-j)), the histograms of the two classes are partially separated, leading to a long time accuracy of 90%, reflecting the residual information in the input norms. Interestingly during the dynamics from the original state to the long time state the distributions cross each other, resulting in a short period of chance performance.

## 6 DISCUSSION

Our work provides the first theoretical understanding of the complete trajectory of gradient-descent learning dynamics of wide DNNs in the presence of small noise, unifying the NTK theory and the NNGP theory as two limits of the same underlying process. While the noise is externally injected in our setup, stochasticity in the machine-learning context may arise from randomness in the data in stochastic gradient descent, making noisy gradient descent a relevant setting in reality (Dalalyan (2017); Noh et al. (2017); Wu et al. (2020); Mignacco & Urbani (2022)). We derive a new kernel, the time-dependent NDK, as a dynamic generalization of the NTK, and provide new insights into learning dynamics in the diffusive learning phase as the learning process explores the solution space. We focus on two particularly interesting phenomena of early stopping and representational drift.

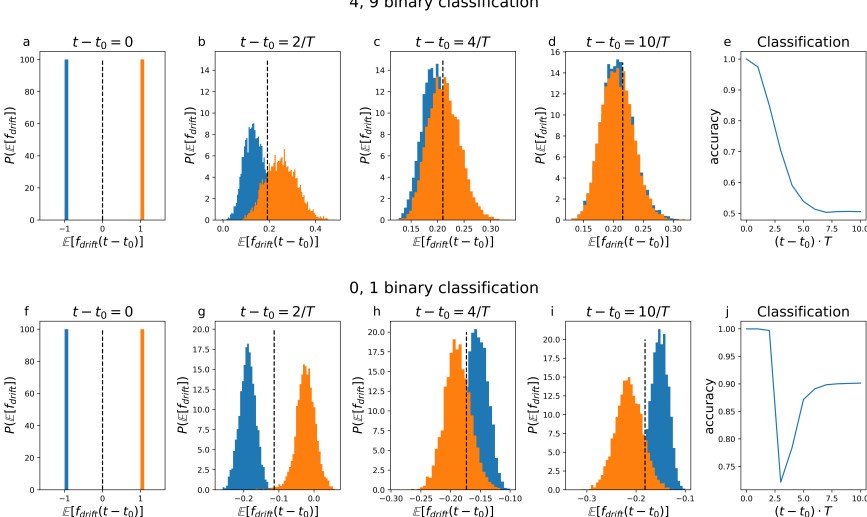

Figure 4: Representational drift with $\mathbf{a}_{t_0}$ fixed at a long-time equilibrium $t_0$. (a-d,f-i) The dynamics of the probability distribution of $\mathbb{E}\left[\mathrm{f}_{\mathrm{drift}}\left(\mathbf{x}, t - t_0\right)\right]$ over the training data, starting with two delta functions at $\pm 1$, and gradually decays in performance when $\mathbf{a}\left(t_0\right)$ and $\mathbf{W}\left(t\right)$ lose alignment. On classification between the digits 0, 1, the norm of the images has enough information to classify them with reasonable success even after complete decorrelation, while on classification between the digits 4,9 the performance is reduced to chance. (e,j) The performance as a function of the time difference from the freezing point $t_0$.

We identify an important parameter $\sigma_0^2/\sigma^2$ characterizing the relative weights amplitude induced by initialization and Bayesian prior regularization, which plays an important role in shaping the trajectories of the predictor. We note that while the results are shown for network with a single output for simplicity, extension to networks with $M$ outputs ($M \sim \mathcal{O}(1)$) is straightforward by simply replacing the $P$ dimensional target output $\mathbf{y}$ to a $P \times M$ dimensional matrix.

In most of our examples, the best performance is achieved after the gradient-driven learning phase, indicating that exploring the solution space improves the network's performance, consistent with empirical findings (Lee et al. (2020)). For some examples, the optimal stopping point occurs during the diffusive phase, before the long-time equilibrium. We stress that our 'early stopping' is 'early' compared to the NNGP equilibrium, and is different from the usual notion of early stopping, which happens in the gradient-driven learning phase (Caruana et al. (2000); Jacot et al. (2018); Advani et al. (2020)). Our theory provides insights into how and when an early stopping point can happen after the network reaches an essentially zero training error.

Our theory for the Langevin dynamics suggests a possible mechanism of representational drift, where the hidden layer weights undergo random diffusion, while the readout weights are continuously realigning to keep performance unchanged, as previously suggested (Rule et al. (2020); Rule & O'Leary (2022)). In our framework, this realignment is due to the presence of a loss-gradient signal. The source of the putative realignment signals in brain circuits is unclear. An alternative hypothesis is that computations in the neuronal circuits are based on features that are invariant to the representational drift (Druckmann & Chklovskii (2012); Kaufman et al. (2014); Rule et al. (2019); Rubin et al. (2019); Deitch et al. (2021); Marks & Goard (2021)). We provide an example of such features and show that performance can be maintained after drift.

So far we have focused on learning in infinitely wide networks in the lazy regime, where the time dependence of the NDK results from random drift in the solution space. Empirical time-dependent NTK is more complex due to feature learning that exists in finite width NNs (Shan & Bordelon (2021); Vyas et al. (2022); Canatar & Pehlevan (2022)) or in an infinite width network with non-lazy regularization (Bordelon & Pehlevan (2022)). Future work aims to extend the theory to the regime where data size is proportional to network width where we expect dynamic kernel renormalization (Li & Sompolinsky (2021; 2022)) and to describe the dynamics of feature learning in non-lazy regularization (Woodworth et al. (2020); Azulay et al. (2021); Flesch et al. (2022)).

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
