(\boldsymbol{\theta}(t) | \mathcal{D}) = \frac{1}{2} |\mathbf{f}_{\text{train}}(t) - \boldsymbol{y}|^2 + \frac{T}{2\sigma^2} |\boldsymbol{\theta}(t)|^2 \tag{3}$$

The first term is the squared error empirical loss (SE loss), and the second term is a regularization term that favors weights with small $L_2$ norm, where $|\boldsymbol{\theta}(t)|^2$ is the sum of the squares of all weights. It is convenient to introduce the temperature parameter (see definition below) $T$ as controlling the relative strength of the regularization, and $\sigma^2$ is the variance of the equilibrium distribution of the Gaussian prior. We consider noisy gradient descent learning dynamics given by continuous-time Langevin dynamics, where the weights of the system start from an i.i.d. Gaussian initial condition with zero mean and variance $\sigma_0^2$, and evolve under gradient descent dynamics with respect to the cost function above with an additive white noise $\boldsymbol{\xi}(t)$:

$$\frac{d}{dt} \boldsymbol{\theta}(t) = -\nabla_{\boldsymbol{\theta}} E(\boldsymbol{\theta}(t)) + \boldsymbol{\xi}(t) \tag{4}$$

where $\boldsymbol{\xi}(t)$ has a white noise statistics $\mathbb{E}\left[ \boldsymbol{\xi}(t) \boldsymbol{\xi}(t')^\top \right] = 2T\boldsymbol{I}\delta(t - t')$, $\mathbb{E}[\boldsymbol{\xi}(t)] = 0$, and $T$ is the temperature controlling the level of noise in the system. As we show below, the additional white noise compared to deterministic gradient descent allows for continued exploration of the solution space after reaching a small training error, and enables the connection from NTK to NNGP theory.

## 2.2 Infinite Width Limit

We are interested in the predictor statistics (in particular the mean predictor) induced by the Langevin dynamics, which can be evaluated analytically in the infinite width where the hidden layer widths are taken to infinity, while the number of training examples $P$ remains finite. For simplicity, we consider all the $N_l$ to be the same for $l = 1, \cdots, L$ and equal to $N$, $N \to \infty$.

SI Sec.A presents a derviation of a path integral formulation of the above Langevin dynamics using a Markov proximal learning framework. Evaluating statistical quantities using these integrals is in general intractable. However, in the infinite width limit, they become tractable, as proven in SI Sec.B.1-B.4. Specifically, the moments of the predictor can be derived from a moment generating function (MGF) $\mathcal{M}[\ell(t)]$, written in the form of a path integral over two auxiliary time-dependent vectors, $\mathbf{u}(t) \in \mathbb{R}^P$ and $\mathbf{v}(t) \in \mathbb{R}^P$. Additionally, $\tilde{\mathbf{u}}(t) = [\mathbf{u}(t), i\ell(t)] \in \mathbb{R}^{P+1}$, where $\ell(t)$ denotes the field coupled to the predictor $f(\mathbf{x}, t)$ on an arbitrary test input vector $\mathbf{x}$. Therefore, the moments of the predictor can be derived by evaluating the derivative of $\mathcal{M}[\ell(t)]$ at $\ell(t) = 0$. In SI Sec.B.4 it is shown that the MGF has the following form,

$$\mathcal{M}[\ell(t)] = \int D\mathbf{u}(t) \int D\mathbf{v}(t) \exp\left( -S[\mathbf{v}(t), \tilde{\mathbf{u}}(t)] \right) \tag{5}$$

$$S[\mathbf{v}(t), \tilde{\mathbf{u}}(t)] = \frac{1}{2} \int_0^\infty dt \int_0^\infty dt' m(t, t') \tilde{\mathbf{u}}^\top(t) \tilde{\mathbf{K}}^L(t, t') \tilde{\mathbf{u}}(t') \tag{6}$$

$$+ \int_0^\infty dt \int_0^t dt' \tilde{\mathbf{u}}(t)^\top \tilde{\mathbf{K}}^{d,L}(t, t') \mathbf{v}(t') + \int_0^\infty dt \mathbf{u}(t)^\top (\mathbf{v}(t) - i\boldsymbol{y})$$

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

}_{\mu,\nu}^{d,L}(t, t') \equiv \tilde{\mathbf{K}}_{\mu,\nu}^{d,L}(t, t')$ and $\mathbf{k}_\mu^{d,L}(t, t') \equiv \tilde{\mathbf{K}}_{P+1,\mu}^{d,L}(t, t')$, respectively. Throughout the paper we will use similar notations for these kernel matrices and vectors (i.e., $\mathbf{K} \in \mathbb{R}^{P \times P}$ for kernel functions applied on training data, and $\mathbf{k} \in \mathbb{R}^P$ for kernel functions applied on test and training data).

## 3 CORRESPONDENCE TO NTK AND NNGP

In this section, we show that our theory recovers known results of the NTK (Jacot et al. (2018)) and NNGP theories (Lee et al. (2018)) in the short- and long-time limit, respectively. We stress that the separation of time scales occurs in the limit of small but nonzero noise controlled by $T$, which is also the relevant limit of a realistic machine-learning scenario.

### 3.1 GRADIENT-DRIVEN PHASE CORRESPONDS TO NTK DYNAMICS

The time dependence of the NDK (Eq.12) is in time scales of $T \cdot t$ (Eqs.9,12), and thus at low $T$ and $t \sim \mathcal{O}(1)$ we can substitute $K^{d,L}(t, t', \mathbf{x}, \mathbf{x}') = K^{d,L}(0, 0, \mathbf{x}, \mathbf{x}')$. In Sec.2.3, we obtain an exact equivalence between the NDK at time zero and the NTK. In this regime, $\mathbf{K}^{d,L}(t, t')$ and $\mathbf{k}^{d,L}(t, t')$ in Eq.15 and Eq.14 become time-independent. By taking the time derivative on both sides, the integral equations Eq.15 and Eq.14 can be transformed into a linear ODE, and solved analytically, leading to the well-known mean predictor in the NTK theory:

$$\mathbb{E}\left[\mathbf{f}\left(\mathbf{x}, t\right)\right] \approx \mathbf{k}_{NTK}^L{}^\top \left[\mathbf{K}_{NTK}^L\right]^{-1} \left(I - \exp\left(-\mathbf{K}_{NTK}^L

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

# A   MARKOV PROXIMAL LEARNING (MPL) FRAMEWORK FOR LEARNING DYNAMICS

We introduce a Markov proximal learning framework for learning dynamics in fully connected deep neural networks (DNNs). This method allows us to construct a dynamical mean field theory for Langevin dynamics in the infinite width limit. We formally write down the moment generating function (MGF) of the predictor. We then use the well-known replica method in statistical physics (Mézard et al. (1987); Franz et al. (1992)), which has also been shown to be a powerful tool for deriving analytical results for learning in NNs (Gardner (1988); Gabrié et al. (2018); Carleo et al. (2019); Bahri et al. (2020); Saglietti & Zdeborová (2022)). We analytically calculate the MGF after averaging over the posterior distribution of the network weights in the infinite width limit, which enables us to compute statistics of the predictor.

## A.1   DEFINITION OF MPL

We consider the network learning dynamics as a Markov proximal process, which is a generalized version of the *deterministic* proximal algorithm (Parikh et al. (2014); Polson et al. (2015)). Deterministic proximal algorithm with $L_2$ regularization is a sequential update rule defined as $\boldsymbol{\theta}_t\left(\boldsymbol{\theta}_{t-1}, \mathcal{D}\right) = \arg\min_{\boldsymbol{\theta}} \left(E\left(\boldsymbol{\theta}|\mathcal{D}\right) + \frac{\lambda}{2}\left|\boldsymbol{\theta} - \boldsymbol{\theta}_{t-1}\right|^2\right)$ where $\lambda$ is a parameter determining the strength of the proximity constraint. This algorithm has been proven to converge to the global minimum for convex cost functions (Teboulle (1997); Drusvyatskiy & Lewis (2018)), and many optimization algorithms widely used in machine learning can be seen as its approximations(Robbins & Monro (1951); Amari (1998); Beck & Teboulle (2003); Bae et al. (2022)). We define a stochastic extension of proximal learning, the Markov proximal learning, through the following transition matrix

$$\mathcal{T}\left(\boldsymbol{\theta}_t|\boldsymbol{\theta}_{t-1}\right) = \frac{1}{Z\left(\boldsymbol{\theta}_{t-1}\right)} \exp\left(-\frac{1}{2}\beta\left(E\left(\boldsymbol{\theta}_t\right) + \frac{\lambda}{2}\left|\boldsymbol{\theta}_t - \boldsymbol{\theta}_{t-1}\right|^2\right)\right) \tag{22}$$

where $Z\left(\boldsymbol{\theta}_{t-1}\right)$ is the single-time partition function, $Z\left(\boldsymbol{\theta}_{t-1}\right) = \int d\boldsymbol{\theta}' \mathcal{T}\left(\boldsymbol{\theta}'|\boldsymbol{\theta}_{t-1}\right)$. $\beta = T^{-1}$ is an inverse temperature parameter characterizing the level of 'uncertainty' and $\beta \to \infty$ limit recovers the deterministic proximal algorithm. We further assume that the initial distribution of $\boldsymbol{\theta}$ is an i.i.d. Gaussian with variance $\sigma_0^2$ and zero mean. Finally, we note that in the large $\lambda$ limit, the difference between $\boldsymbol{\theta}_t$ and $\boldsymbol{\theta}_{t-1}$ is infinitesimal, and $\boldsymbol{\theta}_t$ becomes a smooth function of continuous time, where the time variable is the discrete time divided by $\lambda$.

**Large $\lambda$ limit and Langevin dynamics:**

We show that in the limit of large $\lambda$ and differentiable cost function this algorithm is equivalent to gradient descent with white noise (Langevin dynamics). We define $\delta\boldsymbol{\theta}_t = \boldsymbol{\theta}_t - \boldsymbol{\theta}_{t-1}$. In the limit of large $\lambda$, we can expand the transition matrix around $\delta\boldsymbol{\theta}_t = 0$:

$$\mathcal{T}\left(\delta\boldsymbol{\theta}_t|\boldsymbol{\theta}_{t-1}\right) \approx \left(\frac{\lambda\beta}{4\pi}\right)^{\frac{d}{2}} \exp\left[-\frac{\lambda\beta}{4}\left|\delta\boldsymbol{\theta}_t + \frac{1}{\lambda}\nabla E\left(\boldsymbol{\theta}_{t-1}\right)\right|^2\right] \tag{23}$$

$\delta\boldsymbol{\theta}_t|\boldsymbol{\theta}_{t-1}$ is Gaussian with statistics:

$$\mathbb{E}\left[\delta\boldsymbol{\theta}_t|\boldsymbol{\theta}_{t-1}\right] = -\frac{1}{\lambda}\nabla E\left(\boldsymbol{\theta}_{t-1}\right) \tag{24}$$

$$\text{Var}\left(\delta\boldsymbol{\theta}_t^i \delta\boldsymbol{\theta}_{t'}^j|\boldsymbol{\theta}_{t-1}\right) = \frac{2}{\lambda\beta}\delta_{ij}\delta_{t,t'} \tag{25}$$

which is equivalent to Langevin dynamics in Itô discretization:

$$\delta\boldsymbol{\theta}_t = \left(-\nabla E\left(\boldsymbol{\theta}_{t-1}\right) + \boldsymbol{\xi}_{t-1}\right)dt \tag{26}$$

with

$$\mathbb{E}\left[\boldsymbol{\xi}_t\boldsymbol{\xi}_{t'}^\top\right] = \frac{2T\boldsymbol{I}\delta_{t,t'}}{dt}, \mathbb{E}\left[\boldsymbol{\xi}_t\right] = 0 \tag{27}$$

where $\frac{1}{\lambda} = dt, \beta = \frac{1}{T}$.

# B CALCULATION OF THE MOMENT GENERATION FUNCTION (MGF) AND THE MEAN PREDICTOR

In this section, we start from the MPL framework introduced in Section A.1, and present the detailed derivation of the moment generating function for the predictor statistics, explain the introduction of the auxilliary variables $\mathbf{v}(t)$ and $\mathbf{u}(t)$ in Eq.37, and derive expressions for the mean predictor given by Eq.15.

## B.1 REPLICA CALCULATION OF THE MGF FOR THE PREDICTOR

The transition matrix can be written using the replica method, where

$$
Z^{-1}\left(\boldsymbol{\theta}_{t-1}\right) = \lim_{n\to 0} Z^{n-1}\left(\boldsymbol{\theta}_{t-1}\right) = \lim_{n\to 0}\left(\int d\boldsymbol{\theta}_t \exp\left(-\frac{\beta}{2}\left(E(\boldsymbol{\theta}_t)+\frac{\lambda}{2}\left|\boldsymbol{\theta}_t-\boldsymbol{\theta}_{t-1}\right|^2\right)\right)\right)^{n-1}
$$

$$
= \lim_{n\to 0}\int \prod_{\alpha=1}^{n-1} d\boldsymbol{\theta}_t^\alpha \exp\left(-\frac{\beta}{2}\left(\sum_{\alpha=1}^{n-1} E(\boldsymbol{\theta}_t^\alpha)+\frac{\lambda}{2}\sum_{\alpha=1}^{n-1}\left|\boldsymbol{\theta}_t^\alpha-\boldsymbol{\theta}_{t-1}\right|^2\right)\right)
$$

(28)

therefore we have

$$
\mathcal{T}\left(\boldsymbol{\theta}_t|\boldsymbol{\theta}_{t-1}\right) = \mathcal{T}(\boldsymbol{\theta}_t^n|\boldsymbol{\theta}_{t-1}^n) = \lim_{n\to 0} Z^{n-1}\left(\boldsymbol{\theta}_{t-1}^n\right)\exp\left(-\frac{1}{2}\beta\left(E\left(\boldsymbol{\theta}_t^n\right)+\frac{\lambda}{2}\left|\boldsymbol{\theta}_t^n-\boldsymbol{\theta}_{t-1}^n\right|^2\right)\right)
$$

$$
= \lim_{n\to 0}\int \prod_{\alpha=1}^{n-1} d\boldsymbol{\theta}_t^\alpha \exp\left(-\frac{\beta}{2}\left(\sum_{\alpha=1}^{n} E\left(\boldsymbol{\theta}_t^\alpha\right)+\frac{\lambda}{2}\sum_{\alpha=1}^{n}\left|\boldsymbol{\theta}_t^\alpha-\boldsymbol{\theta}_{t-1}^n\right|^2\right)\right)
$$

(29)

Here $\alpha = 1,\cdots,n-1$ are the 'replicated copies' of the physical variable $\{\boldsymbol{\theta}_\tau^n\}_{\tau=1,\cdots,t} \equiv \{\boldsymbol{\theta}_\tau\}_{\tau=1,\cdots t}$. To calculate the statistics of the dynamical process, we consider the MGF for arbitrary functions of the trajectory $g(\{\boldsymbol{\theta}_\tau^n\}_{\tau=0,\cdots t})$, $\mathcal{M}\left[\ell_t\right] = \mathbb{E}\left[e^{\ell_t g\left(\{\boldsymbol{\theta}_\tau^n\}_{\tau=0\cdots t}\right)}\right]$

$$
\mathcal{M}\left[\ell_t\right] = \prod_{\tau=0}^{\infty}\int d\boldsymbol{\theta}_\tau \left[\prod_{\tau=1}^{\infty}\mathcal{T}\left(\boldsymbol{\theta}_\tau|\boldsymbol{\theta}_{\tau-1}\right)\right] p\left(\boldsymbol{\theta}_0\right)\exp\left(\sum_{t=1}^{\infty}\ell_t g\left(\{\boldsymbol{\theta}_\tau^n\}_{\tau=0,\cdots t}\right)\right)
$$

(30)

$$
= \lim_{n\to 0}\prod_{\alpha=1}^{n}\prod_{\tau=1}^{\infty}\int d\boldsymbol{\theta}_t^\alpha \int d\boldsymbol{\theta}_0^n p\left(\boldsymbol{\theta}_0^n\right)
$$

$$
\exp\left(-\frac{\beta}{2}\sum_{\tau=1}^{\infty}\left(\sum_{\alpha=1}^{n} E\left(\boldsymbol{\theta}_\tau^\alpha\right)+\frac{\lambda}{2}\sum_{\alpha=1}^{n}\left|\boldsymbol{\theta}_\tau^\alpha-\boldsymbol{\theta}_{\tau-1}^n\right|^2\right)+\sum_{t=1}^{\infty}\ell_t g\left(\{\boldsymbol{\theta}_\tau^n\}_{\tau=0,\cdots t}\right)\right)
$$

(31)

We now apply this formalism to the cost function from Sec.2.1:

$$
E\left(\boldsymbol{\theta}_t|\mathcal{D}\right) = \frac{1}{2}\left|\mathbf{f}_{\text{train}}\left(t\right)-\boldsymbol{y}\right|^2 + \frac{T}{2\sigma^2}\left|\boldsymbol{\theta}_t\right|^2
$$

(32)

and the predictor statistics at time $t$, $g(\{\boldsymbol{\theta}_\tau^n\}_{\tau=0,\cdots t}) = \mathbf{f}\left(\mathbf{x},\boldsymbol{\theta}_t^n\right)$, yielding

$$
\mathcal{M}\left[\ell_t\right] = \lim_{n\to 0}\prod_{\alpha=1}^{n}\prod_{\tau=1}^{\infty}\int d\boldsymbol{\theta}_\tau^\alpha \int d\boldsymbol{\theta}_0 \exp\left(-\frac{\beta}{4}\sum_{\tau=1}^{t}\sum_{\alpha=1}^{n}\left|\mathbf{f}_{\text{train}}\left(\boldsymbol{\theta}_t^\alpha\right)-\boldsymbol{y}\right|^2 + \ell_t\mathbf{f}\left(\mathbf{x},\boldsymbol{\theta}_t^n\right)-S_0\left[\boldsymbol{\theta}\right]\right)
$$

(33)

$$
S_0\left[\boldsymbol{\theta}\right] = \frac{1}{4}\sum_{\tau=1}^{\infty}\sum_{\alpha=1}^{n}\left(\sigma^{-2}\left|\boldsymbol{\theta}_\tau^\alpha\right|^2 + \lambda\beta\left|\boldsymbol{\theta}_\tau^\alpha-\boldsymbol{\theta}_{\tau-1}^n\right|^2\right)+\frac{1}{2}\sigma_0^{-2}\left|\boldsymbol{\theta}_0^n\right|^2
$$

(34)

where we define $\mathbf{f}_{\text{train}}\left(t\right) \equiv \left[\mathbf{f}\left(\mathbf{x}^1,t\right),\cdots,\mathbf{f}\left(\mathbf{x}^\mu,t\right)\right] \in \mathbb{R}^P$ a vector contains the predictor on the training dataset, and $\boldsymbol{y} \in \mathbb{R}^P$ as in the main text (Sec.2.1). $S_0\left(\boldsymbol{\theta}\right)$ denotes the Gaussian prior on the parameters including the hidden layer weights and the readout weights.

To perform the integration over $\{\mathbf{a}_\tau^\alpha\}$, we use Hubbard-Stratonovich (H.S.) transformation and introduce a new vector field $\mathbf{v}_\tau^\alpha \in \mathbb{R}^P$

$$\mathcal{M}[\ell_t] = \lim_{n \to 0} \prod_{\alpha=1}^n \prod_{\tau=1}^\infty \int d\boldsymbol{\theta}_\tau^\alpha \int d\mathbf{v}_\tau^\alpha \int d\boldsymbol{\theta}_0 \tag{35}$$

$$\exp\left( -\frac{i\beta}{2} \sum_{\tau=1}^\infty \sum_{\alpha=1}^n \left( \frac{1}{\sqrt{N_L}} \mathbf{f}_{\text{train}}(t) - \boldsymbol{y} \right)^\top \mathbf{v}_\tau^\alpha \right.$$

$$\left. -\frac{\beta}{4} \sum_{\tau=1}^\infty \sum_{\alpha=1}^n |\mathbf{v}_\tau^\alpha|^2 + \ell_t \mathbf{f}(\mathbf{x}, \boldsymbol{\theta}_t^n) - S_0(\boldsymbol{\theta}_\tau^\alpha) \right)$$

**Averaging over the readout weights a:**

We denote the hidden layer weights collectively as $\mathcal{W}_\tau^\alpha = \{\mathbf{W}_\tau^{1,\alpha} \cdots \mathbf{W}_\tau^{L,\alpha}\}$. We integrate over $\mathbf{a}_\tau^\alpha$

$$\mathcal{M}[\ell_\tau] = \lim_{n \to 0} \prod_{\tau=1}^\infty \prod_{\alpha=1}^n \int d\mathbf{v}_\tau^\alpha \int d\mathcal{W}_\tau^\alpha \tag{36}$$

$$\exp\left( -S[\mathbf{v}_\tau^\alpha, \mathcal{W}_\tau^\alpha] - Q[\ell_t, \mathbf{v}_\tau^\alpha, \mathcal{W}_\tau^\alpha] - S_0[\mathcal{W}_\tau^\alpha] \right)$$

$$S[\mathbf{v}_\tau^\alpha, \mathcal{W}_\tau^\alpha] = \frac{\beta}{4} \left( \sum_{\alpha,\beta=1}^n \sum_{\tau=1}^\infty \frac{\beta}{2} \mathbf{v}_\tau^{\alpha\top} m_{\tau,\tau'}^{\alpha\beta} \mathbf{K}_{\tau,\tau'}^{L,\alpha\beta}(\mathcal{W}_\tau^\alpha) \mathbf{v}_{\tau'}^\beta + \sum_{\alpha=1}^n \sum_{\tau=1}^\infty (\mathbf{v}_\tau^\alpha - 2iY)^\top \mathbf{v}_\tau^\alpha \right) \tag{37}$$

and the source term action

$$Q[\ell_t, \mathbf{v}_\tau^\alpha, \mathcal{W}_\tau^\alpha] = i\frac{\beta}{2} \sum_{\alpha=1}^n \sum_{t,\tau=1}^\infty \mathbf{v}_\tau^{\alpha\top} m_{t,\tau}^{\alpha n} \mathbf{k}_{t,\tau}^{L,\alpha n}(\mathcal{W}_\tau^\alpha) \ell_t \tag{38}$$

$$-\frac{1}{2} \sum_{t,t'=1}^\infty m_{t,t'}^{nn} k_{t,t'}^{L,nn}(\mathcal{W}_\tau^n) \ell_t \ell_{t'}$$

Where $m_{\tau,\tau'}^{\alpha\beta}$ is a scalar function independent of the data, and represents the averaging w.r.t. to the replica dependent prior $S_0[\boldsymbol{\theta}_\tau^\alpha]$, such that

$$\mathbb{E}\left[ (\boldsymbol{\theta}_\tau^\alpha)_i (\boldsymbol{\theta}_{\tau'}^\beta)_j \right]_{S_0} = \delta_{ij} m_{\tau,\tau'}^{\alpha\beta}$$

$$m_{\tau,\tau'}^{\alpha\beta} = \begin{cases} m_{\tau,\tau'}^1 = \tilde{\sigma}^2 \left( \tilde{\lambda}^{|\tau-\tau'|} + \gamma \tilde{\lambda}^{\tau+\tau'} \right) & \{\alpha = \beta, \tau = \tau'\} \cup \{\alpha = n, \tau < \tau'\} \cup \{\beta = n, \tau > \tau'\} \\ m_{\tau,\tau'}^0 = \tilde{\sigma}^2 \left( \tilde{\lambda}^2 \tilde{\lambda}^{|\tau-\tau'|} + \gamma \tilde{\lambda}^{\tau+\tau'} \right) & otherwise \end{cases} \tag{39}$$

where we have defined new functions of the parameters for convenience,

$$\tilde{\lambda} = \frac{\lambda}{\lambda + T\sigma^{-2}}, \tilde{\sigma}^2 = \sigma^2 \frac{\lambda + T\sigma^{-2}}{\lambda + \frac{1}{2}T\sigma^{-2}}, \gamma = \frac{\sigma_0^2}{\tilde{\sigma}^2} - 1 \tag{40}$$

The time-dependent and replica-dependent kernel function $K_{\tau,\tau'}^{L,\alpha\beta}(\mathbf{x}, \mathbf{x}')$ is defined as:

$$K_{\tau,\tau'}^{L,\alpha\beta}(\mathbf{x}, \mathbf{x}') = \frac{1}{N_L} \left( \mathbf{x}_\tau^L(\mathbf{x}, \mathcal{W}_\tau^\alpha) \cdot \mathbf{x}_{\tau'}^L(\mathbf{x}', \mathcal{W}_{\tau'}^\beta) \right) \tag{41}$$

And $\mathbf{K}_{\tau,\tau'}^{L,\alpha\beta} \in \mathbb{R}^{P \times P}, \mathbf{k}_{\tau,\tau'}^{L,\alpha\beta} \in \mathbb{R}^P, k_{\tau,\tau'}^{L,\alpha\beta} \in \mathbb{R}$ are given by applying the kernel function on the training data and test data, respectively.

**Averaging over the hidden layer weights $\mathcal{W}$** :

In the infinite width limit, the statistics of $\mathcal{W}$ are dominated by its Gaussian prior (Eq.34) with zero mean and covariance $\langle \mathcal{W}_\tau^\alpha \mathcal{W}_{\tau'}^{\beta\top} \rangle = m_{\tau,\tau'}^{\alpha\beta} \boldsymbol{I}$ .Thus the averaged kernel function $K_{\tau,\tau'}^{L,\alpha\beta}(\mathbf{x},\mathbf{x}')$ (Eq.41) over the prior yields two kinds of statistics for a given pair of time $\{\tau,\tau'\}$ as for $m_{\tau,\tau'}^{\alpha\beta}$, which we denote as $K_{\tau,\tau'}^{1,L}(\mathbf{x},\mathbf{x}')$, and $K_{\tau,\tau'}^{0,L}(\mathbf{x},\mathbf{x}')$ :

$$
K_{\tau,\tau'}^{L,\alpha\beta}(\mathbf{x},\mathbf{x}') = \begin{cases} K_{\tau,\tau'}^{1,L}(\mathbf{x},\mathbf{x}') & \{\alpha=\beta,\tau=\tau'\} \cup \{\alpha=n,\tau<\tau'\} \cup \{\beta=n,\tau>\tau'\} \\ K_{\tau,\tau'}^{0}(\mathbf{x},\mathbf{x}') & otherwise \end{cases} \tag{42}
$$

And they obey the iterative relations:

$$
K_{\tau,\tau'}^{1,L}(\mathbf{x},\mathbf{x}') = F\left(m_{\tau,\tau}^1 K_{\tau,\tau}^{1,L-1}(\mathbf{x},\mathbf{x}), m_{\tau',\tau'}^1 K_{\tau',\tau'}^{1,L-1}(\mathbf{x}',\mathbf{x}'), m_{\tau,\tau'}^1 K_{\tau,\tau'}^{1,L-1}(\mathbf{x},\mathbf{x}')\right) \tag{43}
$$

$$
K_{\tau,\tau'}^{0,L}(\mathbf{x},\mathbf{x}') = F\left(m_{\tau,\tau}^1 K_{\tau,\tau}^{1,L-1}(\mathbf{x},\mathbf{x}), m_{\tau',\tau'}^1 K_{\tau',\tau'}^{1,L-1}(\mathbf{x}',\mathbf{x}'), m_{\tau,\tau'}^0 K_{\tau,\tau'}^{0,L-1}(\mathbf{x},\mathbf{x}')\right) \tag{44}
$$

$$
K^{1,L=0}(\mathbf{x},\mathbf{x}') = K^{0,L=0}(\mathbf{x},\mathbf{x}') = K^{in}(\mathbf{x},\mathbf{x}') \tag{45}
$$

$$
K^{in}(\mathbf{x},\mathbf{x}') = \frac{1}{N_0}\sum_{i=1}^N \mathbf{x}_i \mathbf{x}_i' \tag{46}
$$

where $F\left(\mathbb{E}\left[z^2\right], \mathbb{E}\left[z'^2\right], \mathbb{E}\left[zz'\right]\right)$ is a nonlinear function of the variances of two Gaussian variables $z$ and $z'$ and their covariance, whose form depends on the nonlinearity of the network (Cho & Saul (2009)). As we see in Eqs.43,44 these variances and covariances depend on the kernel functions of the previous layer and on the prior replica-dependent statistics represented by $m_{\tau,\tau'}^{1,0}$.

The MGF can be written as a function of the statistics of one of these kernels, and their difference, which we will denote as $\Delta_{\tau,\tau'}^L(\mathbf{x},\mathbf{x}') = \frac{\lambda\beta}{2}\left(K_{\tau,\tau'}^{1,L}(\mathbf{x},\mathbf{x}') - K_{\tau,\tau'}^{0,L}(\mathbf{x},\mathbf{x}')\right)$. It is useful to define a new kernel, the discrete neural dynamical kernel $K_{\tau,\tau'}^{d,L}(\mathbf{x},\mathbf{x}') = \lim_{n\to 0}\frac{\lambda\beta}{2}\sum_{\alpha=1}^n m_{\tau,\tau'}^{n\beta} K_{\tau,\tau'}^{n\beta,L}(\mathbf{x},\mathbf{x}')$, which controls the dynamics of the mean predictor. It has a simple expression in terms of the kernel $K_{\tau,\tau'}^{0,L}(\mathbf{x},\mathbf{x}')$ and the kernel difference $\Delta_{\tau,\tau'}^L$.

$$
K_{\tau,\tau'}^{d,L}(\mathbf{x},\mathbf{x}') = \begin{cases} 0 & \tau \le \tau' \\ m_{\tau,\tau'}^1 \Delta_{\tau,\tau'}^L(\mathbf{x},\mathbf{x}') + \tilde{\lambda}^{|\tau-\tau'|+1} K_{\tau,\tau'}^{0,L}(\mathbf{x},\mathbf{x}') & \tau > \tau' \end{cases} \tag{47}
$$

We integrate over the replicated hidden layers variables $\mathcal{W}_\tau^\alpha$, which replaces the $\mathcal{W}$ dependent kernels with the averaged kernels. We get an MGF that depends only on the $\mathbf{v}_\tau^\alpha$ variables

$$
\mathcal{M}[\ell_t] = \lim_{n\to 0}\prod_{\alpha=1}^n\prod_{\tau=1}^\infty \int d\mathbf{v}_\tau^\alpha \exp\left(-S(\mathbf{v}_\tau^\alpha) - Q(\ell_t,\mathbf{v}_\tau^\alpha)\right) \tag{48}
$$

$$
S[\mathbf{v}_\tau^\alpha] = \frac{\beta}{4}\sum_{\tau=1}^\infty \left(\frac{\beta}{2}\sum_{\alpha,\beta=1}^n\sum_{\tau'=1}^\infty \mathbf{v}_\tau^{\alpha\top} m_{\tau,\tau'}^0 \mathbf{K}_{\tau,\tau'}^{0,L}\mathbf{v}_\tau^\beta + \frac{2}{\lambda}\sum_{\alpha=1}^n\sum_{\tau'=1}^{t-1}\mathbf{v}_\tau^{\alpha\top}\mathbf{K}_{\tau,\tau'}^{d,L}\mathbf{v}_{\tau'}^n \right. \tag{49}
$$

$$
\left. + \frac{1}{\lambda}\sum_{\alpha=1}^n \mathbf{v}_\tau^{\alpha\top}\mathbf{K}_{\tau,\tau}^{d,L}\mathbf{v}_\tau^\alpha + \sum_{\alpha=1}^n \mathbf{v}_\tau^{\alpha\top}(\mathbf{v}_\tau^\alpha - 2i\boldsymbol{y})\right)
$$

$$
Q[\ell_t,\mathbf{v}_\tau^\alpha] = \frac{i\beta}{2}\sum_{\beta=1}^n\sum_{t,\tau'=1}^\infty \ell_t m_{t,\tau'}^0 \mathbf{k}_{t,\tau'}^{0,L\top}\mathbf{v}_{\tau'}^\beta + \frac{i}{\lambda}\sum_{t,\tau'=1}^t \ell_t \mathbf{k}_{t,\tau'}^{d,L\top}\mathbf{v}_{\tau'}^n
$$

$$
+ \frac{i}{\lambda}\sum_{\beta=1}^n\sum_{t=1}^\infty\sum_{\tau'=t+1}^\infty \ell_t \mathbf{k}_{t,\tau'}^{d,L\top}\mathbf{v}_{\tau'}^\beta - \sum_{t=1}^\infty \frac{1}{2}m_{t,t}^1 \ell_t^2 k_{t,t}^{1,L} \tag{50}
$$

$\mathbf{k}_{t,\tau'}^{d,L}$ in Eq.50 is a $P$-dimensional vector given by applying the kernel function on the test data.

## B.2 INTEGRATE OUT REPLICATED VARIABLES $\mathbf{v}_\tau^\alpha$

We define a new variable $\mathbf{u}_\tau = \frac{\lambda\beta}{2}\sum_{\alpha=1}^n \mathbf{v}_\tau^\alpha$, and integrate out $\mathbf{v}_\tau^{\alpha\neq n}$, we obtain a simpler expression of the MGF (after taking the limit $n \to 0$).

$$\mathcal{M}\left[\ell_t\right] = \prod_{\tau=1}^\infty \int d\mathbf{v}_\tau \int d\mathbf{u}_\tau \exp\left(-S\left[\mathbf{v}_\tau, \mathbf{u}_\tau\right] - Q\left[\ell_\tau, \mathbf{v}_\tau, \mathbf{u}_\tau\right]\right) \tag{51}$$

$$S\left[\mathbf{v}_\tau, \mathbf{u}_\tau\right] = \frac{1}{2\lambda^2}\sum_{\tau,\tau'=1}^\infty \mathbf{u}_\tau^\top \left(m_{\tau,\tau'}^0 \mathbf{K}_{\tau,\tau'}^{0,L} - \frac{2}{\beta}\delta_{\tau,\tau'}\left(\boldsymbol{I} + \frac{1}{\lambda}\mathbf{K}_{\tau,\tau}^{d,L}\right)\right)\mathbf{u}_{\tau'} \tag{52}$$

$$+ \frac{1}{\lambda}\sum_{\tau=1}^\infty \left(\frac{1}{\lambda}\sum_{\tau'=1}^{\tau-1}\mathbf{K}_{\tau,\tau'}^{d,L}\mathbf{v}_{\tau'} + \left(\boldsymbol{I} + \frac{1}{\lambda}\mathbf{K}_{\tau,\tau}^{d,L}\right)\mathbf{v}_\tau - i\boldsymbol{y}\right)^\top \mathbf{u}_\tau$$

$$Q\left[\ell_\tau, \mathbf{v}_\tau, \mathbf{u}_\tau\right] = \frac{i}{\lambda}\sum_{t=1}^\infty \ell_t \left(\sum_{\tau'=1}^\infty m_{t,\tau'}^0 \mathbf{k}_{t,\tau'}^{0,L\top}\mathbf{u}_{\tau'} + \sum_{\tau'=1}^t \mathbf{k}_{t,\tau'}^{d,L\top}\mathbf{v}_{\tau'} + \frac{2}{\lambda\beta}\sum_{\tau'=t+1}^\infty \mathbf{k}_{t,\tau'}^{d,L\top}\mathbf{u}_{\tau'}\right) \tag{53}$$

$$- \sum_{t=1}^\infty \frac{1}{2}\left(\ell_t\right)^2 m_{t,t}^1 k_{t,t}^{1,L}$$

## B.3 DETAILED CALCULATION OF THE MEAN PREDICTOR

To derive the mean predictor we take the derivative of the MGF w.r.t. $\ell_t$:

$$\mathbb{E}\left[\mathbf{f}\left(\mathbf{x}, t\right)\right] = \left.\frac{\partial\mathcal{M}\left(\ell_t\right)}{\partial\ell_t}\right|_{\ell_t=0} \tag{54}$$

which yields

$$\mathbb{E}\left[\mathbf{f}\left(\mathbf{x}, t\right)\right] = \frac{1}{\lambda}\sum_{t'=1}^t \mathbf{k}_{t,t'}^{d,L\top}\mathbb{E}\left[-i\mathbf{v}_{t'}\right] \tag{55}$$

Furthermore, from the H.S. transformation in Eq.35, we can relate $\mathbb{E}\left[\mathbf{v}_\tau\right]$ to the mean predictor on the training data :

$$\mathbb{E}\left[i\mathbf{v}_t\right] = \mathbb{E}\left[\mathbf{f}_{\text{train}}\left(t\right)\right] - \boldsymbol{y} \tag{56}$$

On the other hand we can get the statistics of $i\mathbf{v}_t$ from the MGF in Eq.51.

$$\mathbb{E}\left[\left(\mathbf{f}_{\text{train}}\right)_t\right] = \left(\boldsymbol{I}\lambda + \mathbf{K}_{t,t}^{d,L}\right)^{-1}\sum_{t'=1}^{t-1}\mathbf{K}_{t,t'}^{d,L}\left(\boldsymbol{y} - \mathbb{E}\left[\left(\mathbf{f}_{\text{train}}\right)_{t'}\right]\right) \tag{57}$$

$$\mathbb{E}\left[\mathbf{f}\left(\mathbf{x}, t\right)\right] = \frac{1}{\lambda}\sum_{t'=1}^t \mathbf{k}_{t,t'}^{d,L\top}\left(\boldsymbol{y} - \mathbb{E}\left[\left(\mathbf{f}_{\text{train}}\right)_{t'}\right]\right) \tag{58}$$

where $\mathbf{K}^{d,L}\left(t, t'\right)$ is a $P \times P$ dimensional kernel matrix defined as $\mathbf{K}_{\mu\nu,t,t'}^{d,L} = \mathrm{K}_{t,t'}^{d,L}\left(\mathbf{x}^\mu, \mathbf{x}^\nu\right)$. Now we can compute $\mathbb{E}\left[\mathbf{f}\left(\mathbf{x}, \boldsymbol{\theta}_t\right)\right]$ iteratively by combining Eqs.57,58.

## B.4 LARGE $\lambda$ LIMIT

All the results so far hold for any $T$ and $\lambda$. Now, we consider the limit where the Markov proximal learning algorithm is equivalent to Langevin dynamics in order to get expressions that are relevant to a gradient-descent scenario. We consider large $\lambda$ and $t_{discrete} \sim O\left(\lambda\right)$, and thus define a new continuous-time $t = t_{discrete}/\lambda \sim O\left(1\right)$. In this limit, the parameters defined in Eq.40 become

$$\tilde{\lambda}^{t_{discrete}} = e^{-T\sigma^{-2}t}, \tilde{\sigma}^2 = \sigma^2 \tag{59}$$

Taking the limit of large $\lambda$ of Eq.51 is straightforward, and yields

$$\mathcal{M}\left[\ell\left(t\right)\right] = \int D\mathbf{v}\left(t\right) \int D\mathbf{u}\left(t\right) \exp\left(-S\left[\mathbf{v}\left(t\right), \mathbf{u}\left(t\right)\right] - Q\left[\ell\left(t\right), \mathbf{v}\left(t\right), \mathbf{u}\left(t\right)\right]\right) \tag{60}$$

where

$$S\left[\mathbf{v}\left(t\right), \mathbf{u}\left(t\right)\right] = \frac{1}{2}\int_0^\infty dt \int_0^\infty dt' m\left(t, t'\right) \mathbf{u}^\top\left(t\right) \mathbf{K}^L\left(t, t'\right) \mathbf{u}\left(t'\right) \tag{61}$$

$$+ \int_0^\infty dt \left(\int_0^t dt' \mathbf{K}^{d,L}\left(t, t'\right) \mathbf{v}\left(t'\right) + \mathbf{v}\left(t\right) - i\boldsymbol{y}\right)^\top \mathbf{u}\left(t\right)$$

and the source term action is

$$Q\left[\ell\left(t\right), \mathbf{v}\left(t\right), \mathbf{u}\left(t\right)\right] = i\int_0^\infty dt \int_0^t dt' \left(\mathbf{k}^{d,L}\left(t, t'\right)\right)^\top \mathbf{v}\left(t'\right) \ell\left(t\right) \tag{62}$$

$$+ i\int_0^\infty dt \int_0^\infty dt' m\left(t, t'\right) \left(\mathbf{k}^L\left(t, t'\right)\right)^\top \mathbf{u}\left(t'\right) \ell\left(t\right)$$

$$- \frac{1}{2}\int_0^\infty dt \int_0^\infty dt' m\left(t, t'\right) k^L\left(t, t'\right) \ell\left(t\right) \ell\left(t'\right)$$

The NDK in Eq.47 can be rewritten as

$$K^{d,L}\left(t, t', \mathbf{x}, \mathbf{x}'\right) = m\left(t, t'\right) \Delta^L\left(t, t', \mathbf{x}, \mathbf{x}'\right) + e^{-T\sigma^{-2}\left|t-t'\right|} K^L\left(t, t', \mathbf{x}, \mathbf{x}'\right) \tag{63}$$

with

$$\Delta^L\left(t, t', \mathbf{x}, \mathbf{x}'\right) = \frac{\lambda}{2T}\left(K^{L,1}\left(t, t', \mathbf{x}, \mathbf{x}'\right) - K^{L,0}\left(t, t', \mathbf{x}, \mathbf{x}'\right)\right) \tag{64}$$

$$= K^{d,L-1}\left(t, t', \mathbf{x}, \mathbf{x}'\right) \dot{K}^L\left(t, t', \mathbf{x}, \mathbf{x}'\right)$$

$$m\left(t, t'\right) = \sigma^2 e^{-T\sigma^{-2}\left|t-t'\right|} + \left(\sigma_0^2 - \sigma^2\right) e^{-T\sigma^{-2}\left(t+t'\right)} \tag{65}$$

with the kernels defined in Sec.2.3 in the main text. Here the quantity $m\left(t, t'\right)$ is the continuous time limit of $m_{t,t'}^1$. As defined in Eq.39, it represents the covariance of the prior

$$\mathbb{E}\left[\boldsymbol{\theta}_t^i \boldsymbol{\theta}_{t'}^j\right]_{S_0} = \delta_{ij} m\left(t, t'\right), \mathbb{E}\left[\boldsymbol{\theta}_t^i\right]_{S_0} = 0 \tag{66}$$

.

The above calculation leads to the recursion relation of $\mathbf{K}^{d,L}\left(t, t', \mathbf{x}, \mathbf{x}'\right)$ given in Eq.12 in the main text:

$$K^{d,L}\left(t, t', \mathbf{x}, \mathbf{x}'\right) = m\left(t, t'\right) \mathbf{K}^{d,L-1}\left(t, t', \mathbf{x}, \mathbf{x}'\right) \dot{K}^L\left(t, t', \mathbf{x}, \mathbf{x}'\right) \tag{67}$$

$$+ e^{-T\sigma^{-2}\left|t-t'\right|} K^L\left(t, t', \mathbf{x}, \mathbf{x}'\right)$$

with initial condition

$$K^{d,L=0}\left(t, t', \mathbf{x}, \mathbf{x}'\right) = e^{-T\sigma^{-2}\left|t-t'\right|} K^{in}\left(\mathbf{x}, \mathbf{x}'\right) \tag{68}$$

Where $K^{in}\left(\mathbf{x}, \mathbf{x}'\right)$ was defined in Eq.46. We refer to this continuous time $\mathbf{K}^{d,L}\left(t, t', \mathbf{x}, \mathbf{x}'\right)$ as the neural dynamical kernel (NDK). Note that it follows directly from Eq.67 that

$$K^{d,L}\left(0, 0, \mathbf{x}, \mathbf{x}'\right) = K_{NTK}^L\left(\mathbf{x}, \mathbf{x}'\right). \tag{69}$$

For the mean predictor we use the results from the previous section Eqs.56,57,58, take the large $\lambda$ limit and turn the sums into integrals, we obtain

$$\mathbb{E}\left[\mathbf{f}_{\text{train}}(t)\right] = \int_0^t dt' \mathbf{K}^{d,L}(t,t')\left(\boldsymbol{y} - \mathbb{E}\left[\mathbf{f}_{\text{train}}(t')\right]\right) \tag{70}$$

$$\mathbb{E}\left[\mathbf{f}(\mathbf{x},t)\right] = \int_0^t dt'\left(\mathbf{k}^{d,L}(t,t')\right)^\top\left(\boldsymbol{y} - \mathbb{E}\left[\mathbf{f}_{\text{train}}(t')\right]\right) \tag{71}$$

as given in Eqs.14, 15 in the main text.

## B.5   TEMPORAL CORRELATIONS

Previously we considered the predictor with readout weights $\mathbf{a}_t$ and hidden layer weights $\mathcal{W}_t$ at the same time $t$. To reveal the effects of learning on $\mathcal{W}$ and $\mathbf{a}$ separately, we can consider the temporal correlation between $\mathcal{W}$ and $\mathbf{a}$ at different times:

$$c\left(\mathbf{x},t_0,t\right) \equiv \mathbb{E}\left[\mathbf{f}\left(\mathbf{x},\mathbf{a}_{t_0},\mathcal{W}_t\right)\right] = \mathbb{E}\left[\frac{1}{\sqrt{N_L}}\mathbf{a}_{t_0}\cdot\mathbf{x}_t^L\left(\mathbf{x},\mathcal{W}_t\right)\right] \tag{72}$$

We can derive the MGF of this quantity by replacing $\ell g\left(\{\boldsymbol{\theta}_\tau^n\}_{\tau=1,\cdots,t}\right)$ in Eq.31 by $\ell\left(t_0,t\right)c\left(t_0,t\right)$. For convenience, we split the action into three parts, one that previously appeared in the equal time calculation in Eq.61, and two new parts involving the new source $\ell\left(t_0,t\right)$.

$$\mathcal{M}\left[\ell\left(t_0,t\right)\right] = \int D\mathbf{v}\left(t\right)\int D\mathbf{u}\left(t\right)\exp\left(-S\left[\mathbf{v}\left(t\right),\mathbf{u}\left(t\right)\right]\right. \tag{73}$$
$$\left. -Q_1\left[\ell\left(t_0,t\right),\mathbf{u}\left(t\right)\right] - Q_2\left[\ell\left(t_0,t\right),\mathbf{v}\left(t\right)\right]\right)$$

$$Q_1\left[\ell\left(t_0,t\right),\mathbf{u}\left(t\right)\right] = \int_0^\infty dt_0\int_0^\infty dt\int_0^\infty dt'm\left(t_0,t'\right)\left(\mathbf{k}^L\left(t,t'\right)\right)^\top\mathbf{u}\left(t'\right)\ell\left(t_0,t\right) \tag{74}$$
$$+\frac{1}{2}\int_0^\infty dt\int_0^\infty dt'\int_0^\infty dt_0\int_0^\infty dt_0'm\left(t_0,t_0'\right)k^L\left(t,t'\right)\ell\left(t_0,t\right)\ell\left(t_0',t'\right)$$

$$Q_2\left[\ell\left(t_0,t\right),\mathbf{v}\left(t\right)\right] = \int_0^\infty dt\int_0^\infty dt_0\int_0^{\max(t_0,t)} dt'\ell\left(t_0,t\right)\mathbf{v}^\top\left(t'\right) \tag{75}$$
$$\left(\boldsymbol{\theta}\left(t-t'\right)m\left(t_0,t'\right)\mathbf{k}^{d,L-1}\left(t,t'\right)\dot{\mathbf{k}}^L\left(t,t'\right)\right.$$
$$\left. +\boldsymbol{\theta}\left(t_0-t'\right)e^{-T\sigma^{-2}\left|t_0-t'\right|}\mathbf{k}^L\left(t,t'\right)\right)$$

Using the same approach as in Sec.B.3, we get the statistics of $c\left(\mathbf{x},t_0,t\right)$, which depend on whether $t > t_0$ or vice versa:

$$c\left(\mathbf{x},t_0 < t\right) = e^{T\sigma^{-2}(t-t_0)}\int_0^{t_0} dt'\left(\mathbf{k}^{d,L}(t,t')\right)^\top\left(\boldsymbol{y} - \mathbb{E}\left[\mathbf{f}_{\text{train}}(t')\right]\right) \tag{76}$$
$$+\int_{t_0}^t dt'm\left(t',t_0\right)\left(\mathbf{k}^{d,L-1}(t,t')\dot{\mathbf{k}}^L(t,t')\right)^\top\left(\boldsymbol{y} - \mathbb{E}\left[\mathbf{f}_{\text{train}}(t')\right]\right)$$

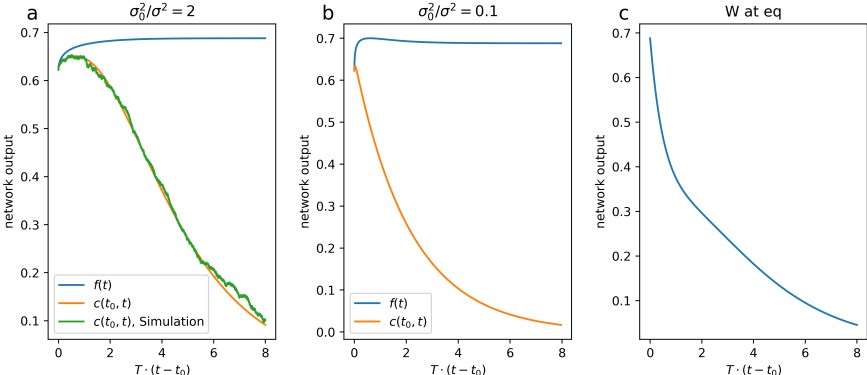

Figure 5: Temporal correlation dynamics for the synthetic dataset. (a-b) Temporal correlations between $\mathbf{a}_{t_0}$ fixed at NTK equilibrium with changing $\mathcal{W}_t$, for different $\sigma_0^2/\sigma^2$ values. In (a), we show remarkable agreement between the theory and network simulations. Interestingly, for larger $\sigma_0^2/\sigma^2$, the temporal correlations closely follow the mean predictor dynamics, meaning the learning of $\mathcal{W}_t$ is dominant in this regime. (b) There is almost exponential decay of the temporal correlations, similar to Fig.11, meaning the effect of learning on $\mathcal{W}_t$ is weak (almost like the representational drift case). (c) $\mathcal{W}_t$ fixed at NNGP equilibrium, while the dynamics of $\mathbf{a}_{t_0}$ continues.

$$c\left(\mathbf{x}, t_0 > t\right) = e^{-T\sigma^{-2}(t_0-t)}\mathbb{E}\left[\mathbf{f}\left(\mathbf{x}, t\right)\right] \tag{77}$$
$$+ \int_t^{t_0} dt' e^{-T\sigma^{-2}(t_0-t)} \left(\mathbf{k}^L\left(t, t'\right)\right)^\top \left(\boldsymbol{y} - \mathbb{E}\left[\mathbf{f}_{\text{train}}\left(t'\right)\right]\right)$$

The kernels are defined in Sec.2.3. $\mathbb{E}\left[\mathbf{f}_{\text{train}}\left(t\right)\right]$ is calculated via the integral equation in Eq.14 in the main text. By definition $c\left(\mathbf{x}, t = t_0\right) = \mathbb{E}\left[\mathbf{f}\left(\mathbf{x}, t\right)\right]$.

Solving the integrals numerically, we find the the ratio $\sigma_0^2/\sigma^2$ plays an important role in the dynamics again. As can be seen in Fig.5 (a), when $\sigma_0^2/\sigma^2$ is large, the temporal correlations follow the predictor for a significant amount of time even though $\mathbf{a}_{t_0}$ is frozen, meaning that the effect of learning on the hidden layer weights $\mathcal{W}_t$ is dominant. Eventually, the decorrelation between $\mathbf{a}_{t_0}$ and $\mathcal{W}_t$ causes a decrease in performance. When $\sigma_0^2/\sigma^2$ is small (Fig.5 (b)), the temporal correlations decrease almost exponentially, hinting that in this regime the effect of learning on the readout weights is dominant. In this case Fig.5 (b) is similar to Fig.11, where there is no external learning signal affecting the hidden layer weights at all.

## C  THE NEURAL DYNAMICAL KERNEL

We focus on the large $\lambda$ limit derived above, and present several examples where the NDK has explicit expressions, and provide proofs of properties of the NDK presented in the main text.

### C.1  LINEAR ACTIVATION:

For linear activation:
$$K^L\left(t, t', \mathbf{x}, \mathbf{x}'\right) = \left(m\left(t, t'\right)\right)^L K^{in}\left(\mathbf{x}, \mathbf{x}'\right) \tag{78}$$

$$\dot{K}^L\left(t, t', \mathbf{x}, \mathbf{x}'\right) = \boldsymbol{I} \tag{79}$$

The recursion relation for the NDK can be solved explicitly, yielding

$$K^{d,L}\left(t, t', \mathbf{x}, \mathbf{x}'\right) = \left(m\left(t, t'\right)\right)^L \left(L+1\right) e^{-T\sigma^{-2}\left|t-t'\right|} K^{in}\left(\mathbf{x}, \mathbf{x}'\right) \tag{80}$$

The NDK of linear activation is proportional to the input kernel $K^{in}(\mathbf{x}, \mathbf{x}')$ regardless of the data. The effect of network depth only changes the magnitude but not the shape of the NDK. As a result, the NNGP and NTK kernels also only differ by their magnitude, and thus the mean predictor at the NNGP and NTK equilibria only differ by $\mathcal{O}(T)$. This suggests that the diffusive phase has very little effect on the mean predictor in the low $T$ regime, as shown in Fig.10.

## C.2   ReLU activation:

For ReLU activation, we define the function $J(\boldsymbol{\theta})$ (Cho & Saul (2009)):

$$J\left(\boldsymbol{\theta}^L\left(t,t',\mathbf{x},\mathbf{x}'\right)\right) = \left(\pi - \boldsymbol{\theta}^L\left(t,t',\mathbf{x},\mathbf{x}'\right)\right)\cos\left(\boldsymbol{\theta}^L\left(t,t',\mathbf{x},\mathbf{x}'\right)\right) + \sin\left(\boldsymbol{\theta}^L\left(t,t',\mathbf{x},\mathbf{x}'\right)\right) \quad (81)$$

where the angle between $\mathbf{x}$ and $\mathbf{x}'$ is given by :

$$\boldsymbol{\theta}^L\left(t,t',\mathbf{x},\mathbf{x}'\right) = \cos^{-1}\left(\frac{m\left(t,t'\right)}{\sqrt{m\left(t,t\right)m\left(t',t'\right)}}\frac{1}{\pi}J\left(\boldsymbol{\theta}^{L-1}\left(t,t',\mathbf{x},\mathbf{x}'\right)\right)\right) \quad (82)$$

$\boldsymbol{\theta}^L\left(t,t',\mathbf{x},\mathbf{x}'\right)$ is defined through a recursion equation, and

$$\boldsymbol{\theta}^{L=0}\left(t,t',\mathbf{x},\mathbf{x}'\right) = \cos^{-1}\left(\frac{m\left(t,t'\right)}{\sqrt{m\left(t,t\right)m\left(t',t'\right)}}\frac{K^{in}\left(\mathbf{x},\mathbf{x}'\right)}{\sqrt{K^{in}(\mathbf{x},\mathbf{x})K^{in}(\mathbf{x}',\mathbf{x}')}}\right) \quad (83)$$

the kernel functions are then given by

$$\dot{K}^L\left(t,t',\mathbf{x},\mathbf{x}'\right) = \frac{1}{2\pi}\left(\pi - \boldsymbol{\theta}^L\left(t,t',\mathbf{x},\mathbf{x}'\right)\right) \quad (84)$$

$$K^L\left(t,t',\mathbf{x},\mathbf{x}'\right) = \frac{\sqrt{K^{in}\left(\mathbf{x},\mathbf{x}\right)K^{in}\left(\mathbf{x}',\mathbf{x}'\right)}}{\pi 2^L}\left(m\left(t,t\right)m\left(t',t'\right)\right)^{L/2}J\left(\boldsymbol{\theta}^{L-1}\left(t,t',\mathbf{x},\mathbf{x}'\right)\right) \quad (85)$$

We obtain an explicit expression for the NDK by plugging these kernels into Eqs.67,68.

## C.3   Error function activation

For error function activation (Williams (1996)):

$$K^L\left(t,t',\mathbf{x},\mathbf{x}'\right)$$
$$= \frac{2}{\pi}\sin^{-1}\left(\frac{2m\left(t,t'\right)K^{L-1}\left(t,t',\mathbf{x},\mathbf{x}'\right)}{\sqrt{\left(1+2m\left(t,t\right)K^{L-1}\left(t,t,\mathbf{x},\mathbf{x}\right)\right)\left(1+2m\left(t',t'\right)K^{L-1}\left(t',t',\mathbf{x}',\mathbf{x}'\right)\right)}}\right) \quad (86)$$

$$\dot{K}^L_{\mu\nu}\left(t,t',\mathbf{x},\mathbf{x}'\right) = \frac{4}{\pi}\left(\left(1+2m\left(t,t\right)K^{L-1}\left(t,t,\mathbf{x},\mathbf{x}\right)\right)\left(1+2m\left(t',t'\right)K^{L-1}\left(t',t',\mathbf{x}',\mathbf{x}'\right)\right)\right.$$
$$\left. -4\left(m\left(t,t'\right)K^{L-1}\left(t,t',\mathbf{x},\mathbf{x}'\right)\right)^2\right)^{-1/2} \quad (87)$$

Again we can obtain an explicit expression for the NDK by plugging these kernels into Eqs.67,68.

## C.4   Long-time behavior of the NDK

We define the long-time limit as $t,t' \to \infty, t-t' \sim \mathcal{O}\left(T^{-1}\right)$. In this limit the statistics of $\mathcal{W}$ w.r.t. the prior becomes only a function of the time difference:

$$\mathbb{E}\left[\mathcal{W}_t\mathcal{W}_{t'}^{\top}\right] = \sigma^2 e^{-T\sigma^{-2}|t-t'|} = m\left(|t-t'|\right) \quad (88)$$

And thus, the kernels defined above will only be functions of the time difference. We look at the time derivative of the kernel (w.l.o.g. we assume $t > t'$), which can be obtained with a chain rule:

$$\frac{d}{dt'}K^L\left(t-t',\mathbf{x},\mathbf{x}'\right) = \dot{K}^L\left(t-t',\mathbf{x},\mathbf{x}'\right)\frac{d}{dt'}\left(K^{L-1}\left(t-t',\mathbf{x},\mathbf{x}'\right)m\left(t-t'\right)\right) \quad (89)$$

We prove by induction:

$$\frac{1}{T}\frac{d}{dt'}\left(m\left(t-t'\right)K^{L}\left(t-t',\mathbf{x},\mathbf{x}'\right)\right)=K^{d,L}\left(t-t',\mathbf{x},\mathbf{x}'\right) \tag{90}$$

The induction basis for $L=0$ is trivial. For arbitrary $L+1$:

$$\frac{1}{T}\frac{d}{dt'}\left(m\left(t-t'\right)K^{L+1}\left(t-t',\mathbf{x},\mathbf{x}'\right)\right) \tag{91}$$

$$= m\left(t-t'\right)\dot{K}^{L+1}\left(t-t',\mathbf{x},\mathbf{x}'\right)\frac{1}{T}\frac{d}{dt'}\left(K^{L}\left(t-t',\mathbf{x},\mathbf{x}'\right)m\left(t-t'\right)\right)$$

$$+ e^{-T\sigma^{-2}\left(t-t'\right)}K^{L+1}\left(t-t',\mathbf{x},\mathbf{x}'\right) \tag{92}$$

And using the induction assumption we get:

$$\frac{1}{T}\frac{d}{dt'}\left(m\left(t-t'\right)K^{L+1}\left(t-t',\mathbf{x},\mathbf{x}'\right)\right) = m\left(t-t'\right)\dot{K}^{L+1}\left(t-t',\mathbf{x},\mathbf{x}'\right)K^{d,L}\left(t-t',\mathbf{x},\mathbf{x}'\right)$$

$$+ e^{-T\sigma^{-2}\left(t-t'\right)}K^{L+1}\left(t-t',\mathbf{x},\mathbf{x}'\right) \tag{93}$$

Which is the expression for $K^{d,L+1}\left(t-t'\right)$. Using this identity, we can get a simple expression for the integral over $K^{d,L}\left(t-t'\right)$ at long times:

$$\lim_{t\to\infty}\int_{0}^{t}dt'K^{d,L}\left(t-t',\mathbf{x},\mathbf{x}'\right)=\frac{\sigma^2}{T}K_{GP}\left(\mathbf{x},\mathbf{x}'\right) \tag{94}$$

As a result, taking the limit of $t\to\infty$ on both sides of Eq.14, we have

$$\lim_{t\to\infty}\mathbb{E}\left[\mathbf{f}_{\text{train}}\left(t\right)\right]=\left(\lim_{t\to\infty}\int_{0}^{t}dt'\mathbf{K}^{d,L}\left(t,t'\right)\right)\left(\boldsymbol{y}-\lim_{t\to\infty}\mathbb{E}\left[\mathbf{f}_{\text{train}}\left(t'\right)\right]\right)$$

$$\lim_{t\to\infty}\mathbb{E}\left[\mathbf{f}_{\text{train}}(t)\right]=\mathrm{K}_{GP}^{\top}\left(\mathrm{K}_{GP}+\sigma^{-2}T\boldsymbol{I}\right)^{-1}\boldsymbol{y} \tag{95}$$

We then take $t\to\infty$ on both sides of Eq.15 and plug in $\lim_{t\to\infty}\mathbb{E}\left[\mathbf{f}_{\text{train}}(t)\right]$ to obtain

$$\lim_{t\to\infty}\mathbb{E}\left[\mathrm{f}(\mathbf{x},t)\right]=\frac{\sigma^2}{T}\mathrm{k}_{GP}^{L}(\mathbf{x})^{\top}(\boldsymbol{y}-\lim_{t\to\infty}\mathbb{E}\left[f_{\text{train}}(t)\right])$$

$$= \left(\mathbf{k}_{GP}^{L}\left(\mathbf{x}\right)\right)^{\top}\left(\boldsymbol{I}T\sigma^{-2}+\mathbf{K}_{GP}^{L}\right)^{-1}\boldsymbol{y} \tag{96}$$

which corresponds to Eq.18 in the main text.

## C.5 NDK AS A GENERALIZED TWO-TIME NTK

In Eq.7 in the main text, we claimed that the NDK has the following interpretation as a generalized two-time NTK

$$K^{d,L}\left(t,t',\mathbf{x},\mathbf{x}'\right)=e^{-T\sigma^{-2}\left|t-t'\right|}\mathbb{E}\left[\nabla_{\boldsymbol{\theta}_t}\mathrm{f}\left(\mathbf{x},\boldsymbol{\theta}_t\right)\cdot\nabla_{\boldsymbol{\theta}_{t'}}\mathrm{f}\left(\mathbf{x}',\boldsymbol{\theta}_{t'}\right)\right]_{S_0}\quad t\ge t' \tag{97}$$

where $\mathbb{E}\left[\cdot\right]_{S_0}$ denotes averaging w.r.t. the prior distribution of the parameters $\boldsymbol{\theta}$, with the statistics defined in Eq.9.

Now we provide formal proof.

We separate $\nabla_{\boldsymbol{\theta}_t}\mathrm{f}\left(\mathbf{x},\boldsymbol{\theta}_t\right)$ into two parts including the derivative w.r.t. the readout weights $a_t$ and the hidden layer weights $\mathcal{W}_t$

**Derivative w.r.t. the readout weights:**

$$\mathbb{E}\left[\partial_{\mathbf{a}_t}\mathrm{f}\left(\mathbf{x},\boldsymbol{\theta}_t\right)\cdot\partial_{\mathbf{a}_{t'}}\mathrm{f}\left(\mathbf{x},\boldsymbol{\theta}_{t'}\right)\right]_{S_0}=K^{L}\left(t,t',\mathbf{x},\mathbf{x}'\right) \tag{98}$$

**Derivative w.r.t. the hidden layer weights:**

We have

$$\partial_{\mathbf{W}_t^l} \mathbf{x}_t^L \left( \mathbf{x}, \mathcal{W}_t \right) = \frac{1}{\sqrt{N_{L-1} \cdots N_{l-1}}} \Pi_{k=l+1}^L \left[ \phi' \left( z_t^k \right) \mathbf{W}_t^k \right] \phi' \left( z_t^l \right) \mathbf{x}_t^{l-1} \tag{99}$$

and

$$\mathbb{E} \left[ \partial_{\mathbf{W}_t^l} \mathrm{f} \left( \mathbf{x}, \boldsymbol{\theta}_t \right) \cdot \partial_{\mathbf{W}_{t'}^l} \mathrm{f} \left( \mathbf{x}, \boldsymbol{\theta}_{t'} \right) \right]_{S_0}$$
$$= \mathbb{E} \left[ N_L^{-1} \mathbf{a}_t \cdot \mathbf{a}_{t'} \right] \left( \Pi_{k=l+1}^L \mathbb{E} \left[ N_k^{-1} N_{k-1}^{-1} \mathbf{W}_t^k \cdot \mathbf{W}_{t'}^k \right] \right) \left( \Pi_{k=l}^L \dot{K}^k \left( t, t', \mathbf{x}, \mathbf{x}' \right) \right) K^{l-1} \left( t, t', \mathbf{x}, \mathbf{x}' \right)$$
$$= m \left( t, t' \right)^{L-l+1} \left( \Pi_{k=l}^L \dot{K}^k \left( t, t', \mathbf{x}, \mathbf{x}' \right) \right) K^{l-1} \left( t, t', \mathbf{x}, \mathbf{x}' \right) \tag{100}$$

To leading order in $N_l$ the averages over $\mathbf{a}$ and $\mathcal{W}$ can be performed separately for each layer, and are dominated by their prior, where each element of the weights is an independent Gaussian given by Eq.34. The term $m \left( t, t' \right)$ comes from the covariance of the priors in $\mathcal{W}$ and $\mathbf{a}$, since there are a total of $L - l$ layers of $\mathcal{W}$ and one layer of $\mathbf{a}$, we have $m \left( t, t' \right)^{L-l+1}$. The kernel $\dot{K}^k \left( t, t', \mathbf{x}, \mathbf{x}' \right)$ comes from the inner product between $\phi' \left( z_t^k \right)$ and $\phi' \left( z_{t'}^k \right)$, and the kernel $K^{l-1} \left( t, t', \mathbf{x}, \mathbf{x}' \right)$ comes from the inner product between $\mathbf{x}_t^{l-1}$ and $\mathbf{x}_{t'}^{l-1}$.

Using proof by induction as for the NTK (Jacot et al. (2018)), we obtain

$$\mathbb{E} \left[ \partial_{\mathcal{W}_t} \mathrm{f} \left( \mathbf{x}, \boldsymbol{\theta}_t \right) \cdot \partial_{\mathcal{W}_{t'}} \mathrm{f} \left( \mathbf{x}, \boldsymbol{\theta}_{t'} \right) \right]_{S_0} = e^{T\sigma^{-2} \left| t - t' \right|} m \left( t, t' \right) \dot{K}^L \left( t, t', \mathbf{x}, \mathbf{x}' \right) K^{d, L-1} \left( t, t', \mathbf{x}, \mathbf{x}' \right) \tag{101}$$

Combine Eq.101 with Eq.98 and with the definition of $K^{d,L} \left( t, t', \mathbf{x}, \mathbf{x}' \right)$ in Eq.67, we have

$$e^{-T\sigma^{-2} \left| t - t' \right|} \mathbb{E} \left[ \nabla_{\boldsymbol{\theta}_t} \mathrm{f} \left( \mathbf{x}, \boldsymbol{\theta}_t \right) \cdot \nabla_{\boldsymbol{\theta}_{t'}} \mathrm{f} \left( \mathbf{x}', \boldsymbol{\theta}_{t'} \right) \right]_{S_0} = K^{d,L} \left( t, t', \mathbf{x}, \mathbf{x}' \right) \tag{102}$$

## D  REPRESENTATIONAL DRIFT

To capture the phenomenon of representational drift, we consider the case where the learning signal stops at some time $t_0$, while the hidden layers continue to drift according to the dynamics of the prior. If all the weights of the system are allowed to drift, the performance of the mean predictor will deteriorate to chance, thus we consider stable readout weights fixed at the end time of learning $t_0$. This scenario can be theoretically evaluated using similar techniques to Sec.B.1 , leading to the following equation for the network output:

$$\mathbb{E} \left[ f_{\mathrm{drift}} \left( \mathbf{x}, t, t_0 \right) \right] = \int_0^{t_0} \left( \mathbf{k}^{d,L} \left( t, t' \right) \right)^\top \left( \boldsymbol{y} - \mathbb{E} \left[ \mathbf{f}_{\mathrm{train}} \left( t' \right) \right] \right) \tag{103}$$

We see here that if $t_0 = t$ it naturally recovers the full mean predictor. It is interesting to look at the limit where the freeze time $t_0$ is at NNGP equilibrium, where the network has finished its dynamics completely. In this case, the expression can be simplified due to the long-time identity of the NDK (Eq.17 in the main text).

$$\mathbb{E} \left[ f_{\mathrm{drift}} \left( t - t_0 \right) \right] = \left( \mathbf{k}^L \left( t - t_0 \right) \right)^\top \left( \boldsymbol{I} T \sigma^{-2} + \mathrm{K}_{GP}^L \right)^{-1} \boldsymbol{y} \tag{104}$$

which has a simple meaning of two samples of hidden layer weights from different times at equilibrium. Even at long time differences, the network performance does not decrease to chance, but reaches a new static state.

$$\lim_{t - t_0 \to \infty} \mathbb{E} \left[ f_{\mathrm{drift}} \left( t - t_0 \right) \right] \to \left( \mathbf{k}_{mean}^L \left( \mathbf{x} \right) \right)^\top \left( \boldsymbol{I} T \sigma^{-2} + \mathrm{K}_{GP}^L \right)^{-1} \boldsymbol{y} \tag{105}$$

We can assess the network's ability to separate classes in a binary classification task by using a linear classifier between the two distributions of outputs (Duda et al. (2000)).

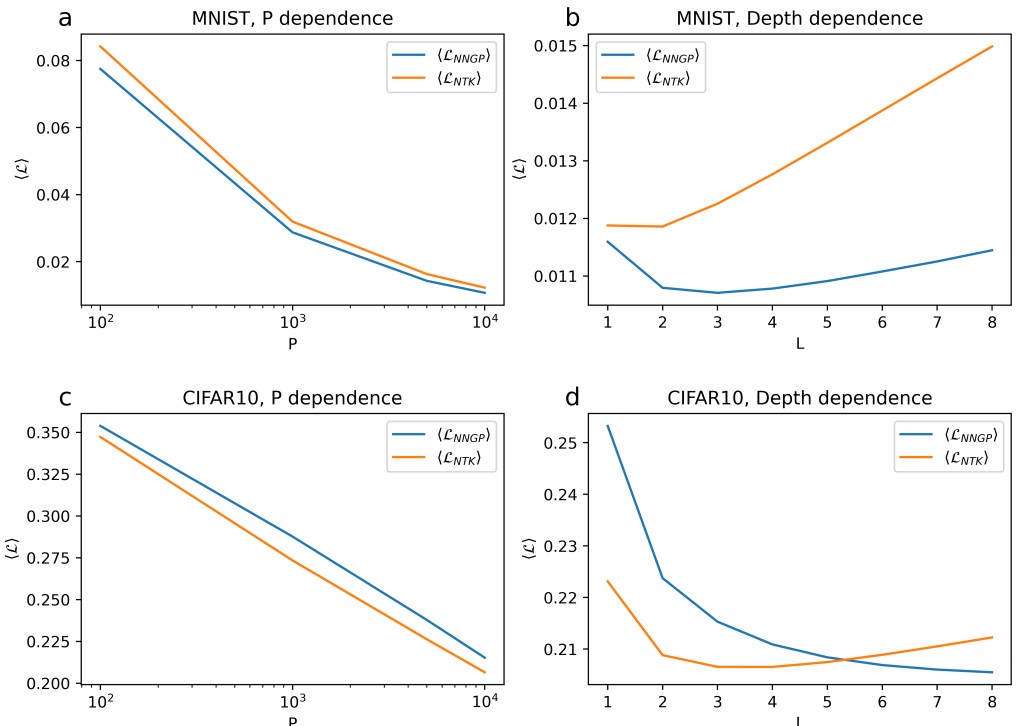

Figure 6: Comparison between NTK and NNGP equilibria, in fully connected DNNs with ReLU activation function.(a-d) The average MSE loss per test example in binary classification tasks of (a,b) MNIST dataset and (c,d) CIFAR10 dataset averaged over all class pairs. We present the results as a function of the number of training examples P (at a constant depth $L = 3$) (a,c), and as a function of depth (at constant $P = 10^4$) (b,d)

### D.1 NTK AND NNGP EQUILIBRIA

The NTK and NNGP equilibria mark the initial and final points for the dynamics of the diffusive phase. An interesting question is how different these two equilibria are. In general, the answer depends on the data and the network architecture (Lee et al. (2020)). In Fig.6 we show that in these tasks deeper networks tend to favor the NNGP equilibrium compared with NTK. On the other hand, increasing the size of the training set has a similar effect on both equilibria.

## E DETAILS OF THE SIMULATIONS

### E.1 SYNTHETIC DATA

We consider $P$ normalized and orthogonal input data vectors $\mathbf{x} \in \mathbb{R}^{N_0}$, such that $\mathbf{K}^{in}_{\mu\nu} = \frac{1}{N_0}\mathbf{x}^\mu \cdot \mathbf{x}^\nu = \delta_{\mu\nu}$. The labels of the data point are $\pm 1$ with equal probability. We consider a test point, which has partial overlap with one of the input vectors, and is orthogonal to all others, w.l.o.g. we assume that the test point is overlapping with the first input vector with label $+1$, such that $\frac{1}{N_0}\mathbf{x}^{test} \cdot \mathbf{x}^\mu = O_{test}\delta_{\mu,1}$, $\frac{1}{N_0}\mathbf{x}^{test} \cdot \mathbf{x}^{test} = 1$, and $y^1 = +1$. In our simulations we set $O_{test} = \frac{3}{4}$, which maximizes the difference between NNGP and NTK equilibria. For this setup, we can represent the kernels by a few scalar functions:

$$\mathbf{K}^{d,L}_{\mu\nu}(t,t') = \mathbf{k}^{d,L}_{\text{offdiag}}(t,t')(1-\delta_{\mu\nu}) + \delta_{\mu\nu}\mathbf{k}^{d,L}_{\text{diag}}(t,t') \tag{106}$$

$$\mathbf{k}^{d,L}_\mu(t,t') = \mathbf{k}^{d,L}_{\text{offdiag}}(t,t')(1-\delta_{\mu,1}) + \delta_{\mu,1}\mathbf{k}^{d,L}_{\text{test}}(t,t') \tag{107}$$

Here $\mathbf{k}_{\text{offdiag}}^{d,L}(t,t')$ and $\mathbf{k}_{\text{diag}}^{d,L}(t,t')$ are off-diagonal and diagonal elements of the kernel matrix $\mathbf{K}^{d,L}(t,t')$, they are scalar functions of time, $\mathbf{k}_{\text{test}}^{d,L}(t,t')$ denotes the first element of the vector $\mathbf{k}(t,t')$, and is also a scalar function of both time and the parameter $O_{test}$. $\mathbf{K}_{\mu\nu}^{L}(t,t')$ and $\dot{\mathbf{K}}_{\mu\nu}^{L}(t,t')$ have the same structure.

Because of the symmetry of this toy model, $\mathbf{f}_{\text{train}}(t)$ takes the same value across all training points with the same label and takes the negative value for training points with the opposite label, and thus can be reduced to a scalar. We consider $\mathbf{f}_{\text{train}}(t)$ on training points with label $+1$. We can transform the vector integral equation into a scalar one, depending only on known scalar functions:

$$\mathbf{f}_{\text{train}}(t) = \int_0^t dt' \left[ \left( \mathbf{k}_{\text{diag}}^{d,L}(t,t') - \mathbf{k}_{\text{diag}}^{d,L}(t,t') \right) (1 - \mathbf{f}_{train}(t')) \right] \tag{108}$$

$$\mathbb{E}\left[ \mathbf{f}(\mathbf{x}, \boldsymbol{\theta}_t) \right] = \int_0^t dt' \left( \mathbf{k}_{\text{test}}^{d,L}(t,t') - \mathbf{k}_{\text{offdiag}}^{d,L}(t,t') \right) (1 - \mathbf{f}_{train}(t')) \tag{109}$$

In this model the theoretical results do not depend on $P, N_0$. For Fig.1, we vary $T$ and $\sigma, \sigma_0$ according to the legend, while keeping $dt = 0.1$. For all other simulations presented, we use $T = 0.001, dt = 0.1$, with total time $t = 10000 = 10/T$, $\sigma = 1$, while $\sigma_0$ varies depending on $\left( \frac{\sigma_0}{\sigma} \right)^2$ that is presented in the plot.

### E.2  MNIST

We consider a digit binary classification task (Deng (2012)), where one type of input is with label $+1$ and the other $-1$. In our simulations we take digits 1 and 0 as the two classes. We take 50 examples from each class, flatten the image into a vector and normalize the data. The test is a previously unseen example from the class $+1$ to make the comparison with other data sets easy (same with the synthetic data and CIFAR10). The examples in the figures are chosen for a large difference between NTK and NNGP equilibria while the error is relatively small. In Fig.2(e-g) example 25910 from MNIST data set is presented, while in Fig.8 examples 50396 (example 2) and 30508 (example 3) are presented. We used $T = 0.01, dt = 0.01$, with total time $t = 1000 = 10/T$. In the simulations presented $\sigma = 1$, while $\sigma_0$ varies depending on $\left( \frac{\sigma_0}{\sigma} \right)^2$ that is presented in the plot.

### E.3  CIFAR10

We consider an image binary classification task (Krizhevsky et al. (2014)), where one class of input is with label $+1$ and the other $-1$. In our simulations we take images of cats and dogs as the two classes. We take 50 examples from each class, flatten the image (including channels) into a vector and normalize the data. The test was a previously unseen example from the class $+1$. The examples in the figures are chosen for a large difference between NTK and NNGP equilibria while the error compared to the true label is relatively small. In Fig.2(h) and in Fig.9 example 4484 from CIFAR10 data set is presented (example 1), while in Fig.9 examples 3287 (example 2), 5430 (example 3) and 6433 (example 4) are presented. We used $T = 0.01, dt = 0.01$, with total time $t = 1000 = 10/T$, and $\sigma = 1$, $\sigma_0$ varies depending on $\left( \frac{\sigma_0}{\sigma} \right)^2$ that is presented in the plot.

### E.4  LANGEVIN DYNAMICS

To check the validity of the theory we performed simulations with Langevin dynamics in a network with $L = 1$, the network is trained under the dynamics given by Eq.26 with $lr = dt = 0.01$, $T = 0.001$, with total time $t = 10000 = 10/T$ on the synthetic data introduced in SI E.1. Simulations shown in Figs.2(a) and Fig.7 are done with $P = 2, N_0 = 100$, and hidden layer width $N = 1000$, $\sigma_0^2/\sigma^2 = 1, 2$ as indicated in the figure captions. Results are averaged over 5000 different initializations and realizations of noise. In the representational drift predictor simulations, at time $t_0$ the loss changes to contain only the prior part, as presented in Eq.34. The network output was calculated with the hidden layer weights at time $t$ with the readout weights at time $t_0$.

# F ADDITIONAL NUMERICAL RESULTS

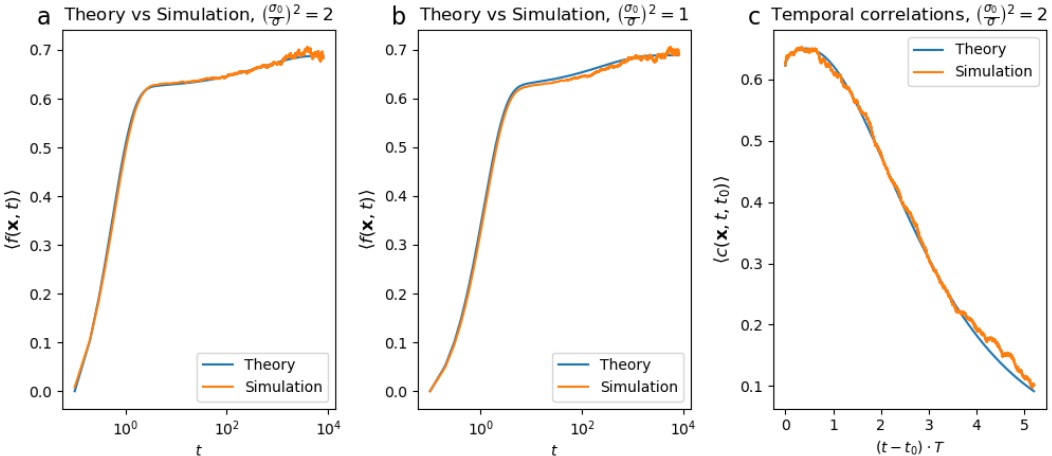

Figure 7: Theory and network simulations of the synthetic data set.(a-b) Theory and simulation of the mean predictor, for different values of $(\sigma_0/\sigma)^2$, with time in log scale due to the large difference in time scales of the two learning phases. (c) Theory and simulation of the temporal correlations of $\mathbf{a}_{t_0}$ at NTK equilibrium with $\mathcal{W}_t$.

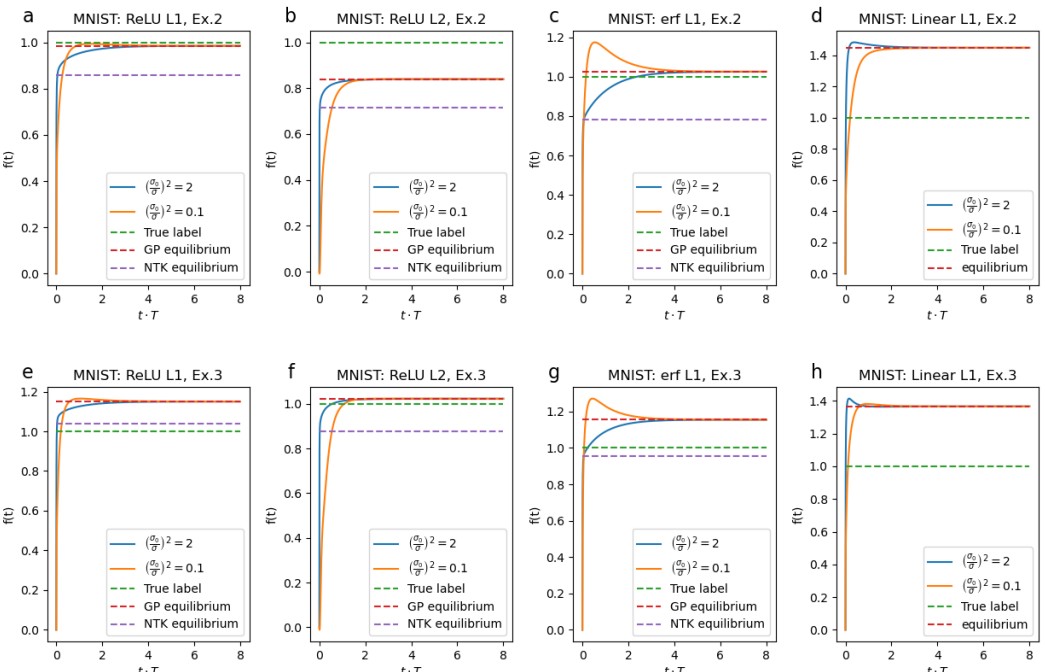

Figure 8: More test examples from MNIST dataset (Deng (2012)), for ReLU ($L = 1, L = 2$), erf and linear activation functions, with different $(\sigma_0/\sigma)^2$ values (a-d) Example 2, with NNGP performance better than NTK. (e-h) Example 3. Interestingly, NTK performance is better than NNGP for ReLU and erf $L = 1$, but is worse for ReLU $L = 2$.

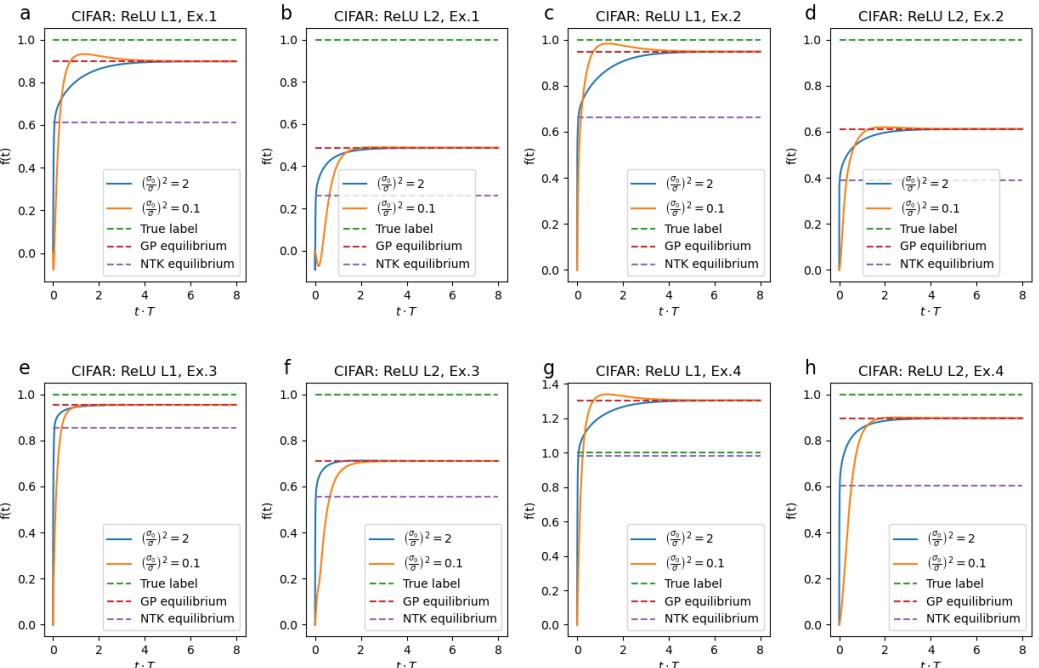

Figure 9: More test examples from CIFAR10 dataset (Krizhevsky et al. (2014)), for ReLU activation function ($L = 1, L = 2$), with different $(\sigma_0/\sigma)^2$ values (a-f) Examples 1,2,3, with NNGP performance better than NTK . (g-h) Example 4. Interestingly, NTK performance is better than NNGP for ReLU $L = 1$, but is worse for ReLU $L = 2$.

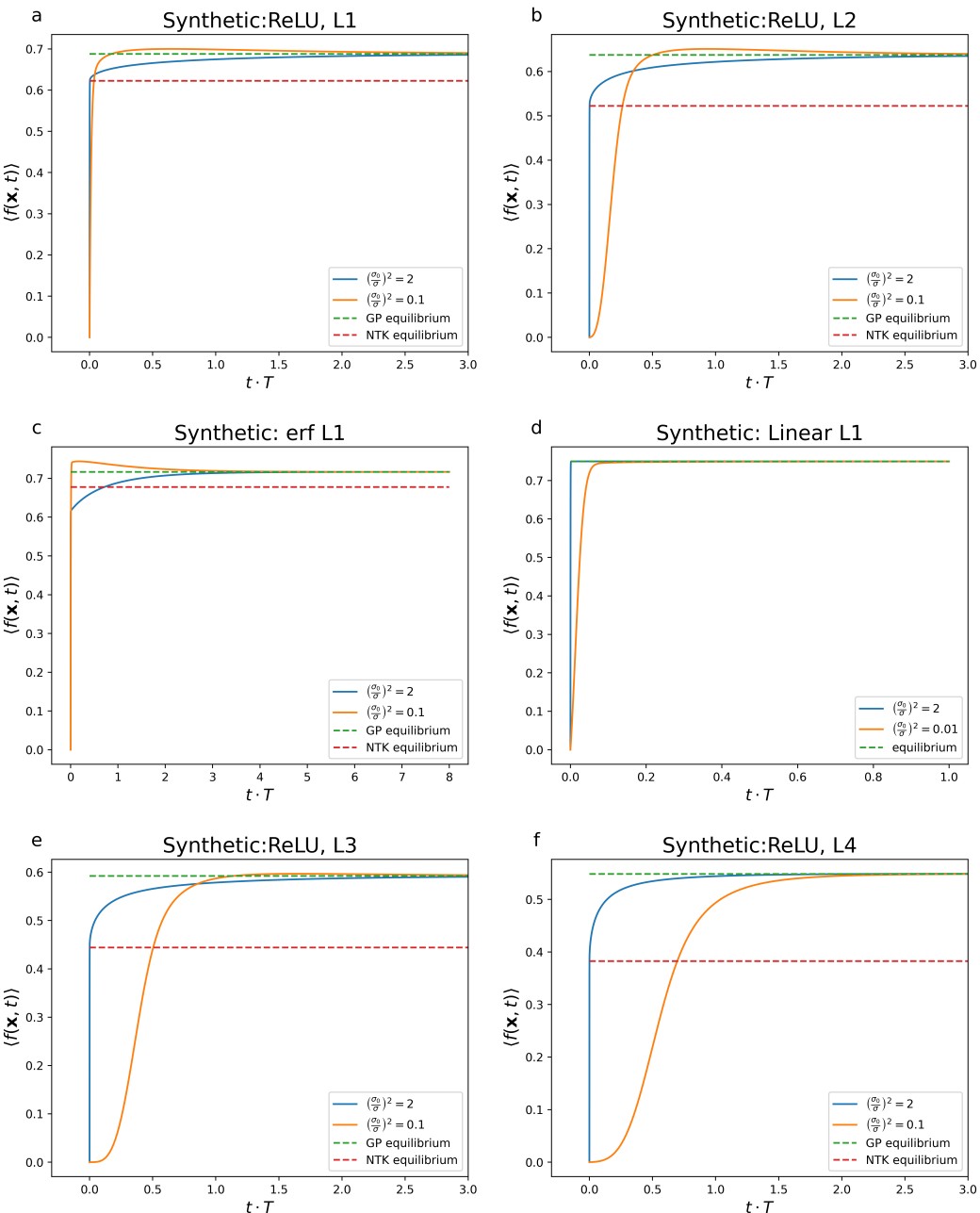

Figure 10: Examples of dynamics for different nonlinearities and depth in the synthetic dataset. (a,b,e,f) ReLU activation function for $L = 1, 2, 3, 4$, with different $(\sigma_0/\sigma)^2$ values. (c) erf activation function with different $(\sigma_0/\sigma)^2$ values. (d) Linear activation function, with different $(\sigma_0/\sigma)^2$ values. We see that with linear activation the system reaches equilibrium in a time shorter than 1/T, during the gradient-driven phase.

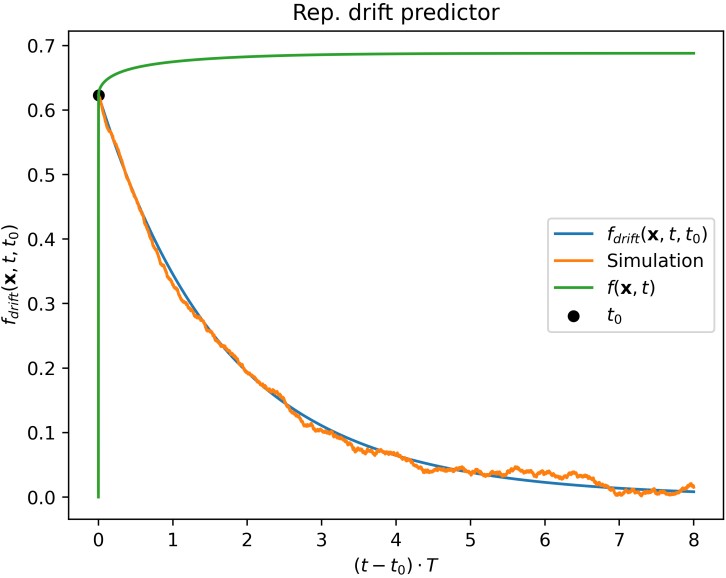

Figure 11: The dynamics of the predictor with no external learning signal and readouts weights frozen at NTK equilibrium ($t_0$), for the synthetic dataset. We see an approximately exponential decay to chance level performance with time scale of $t \sim 1/T$.