# OpenReview forum: "Connecting NTK and NNGP: A Unified Theoretical Framework for Neural Network Learning Dynamics in the Kernel Regime"
_ICLR.cc/2024/Conference — Submitted to ICLR 2024_

### Official Review · Reviewer_Bpa7 · 2023-10-27

**Soundness:** 4 excellent
**Presentation:** 3 good
**Contribution:** 3 good
**Rating:** 8
**Confidence:** 4

**Summary:**

The submission introduces a generalised kernel - Neural Dynamic Kernel (NDK)  for stochastic gradient for very wide (infinite) neural networks. The Neural Tangent Kernel (NTK) and Neural Network Gaussian Process Kernel (NNGP) are two different timescale limits of the NDK - the NTK on the early training as deterministic gradient driven and the later driven by diffusion.

The NDK kernel different from the earlier Kernels as it is explicitly time dependent, and allows, in the short time scale and long time scale limits two different,  NTK (deterministic)  and NNGP (diffusive), like behaviours - the main contribution of the paper.

The argumentation in the appendixes is thorough, and the trial cases support the conclusions.

**Strengths:**

Clear representation, with excellent narrative in the appendixes. These support the main text that concentrates nicely on the main line of thought. Some results are surprising like the different qualitative behaviours of the kernels in the one layer and two layer cases.

**Weaknesses:**

The meaning of  time dependence of the NDK kernel would need discussion.  Already the learning rate of the stochastic gradient is tuned and this is essential for the minimising algorithm to work. Now, two different domains arise when the gradient is big and one is far for the optimal point - it is good to move with big steps down the slope. At the bottom, the finite time step size, in gradient decent, will lead to diffusive Brownian motion around the minimum until the learning rate shrinks to zero. This behaviour is more a property of the stochastic gradient than a property of the neural networks themselves.

As discussed in the manuscript, the practical trials of the Kernel have to be performed (like the Langevin equation) with discretised time. The discretisation errors will lead to extra diffusive contribution and to the above description - unless a hybrid Monte Carlo is used where the evolution of differential equation is proven to be ergodic and reversible. Hence, we cannot easily distinguish the root cause of the phenomenon. There is no clear indication if the behaviours of the Kernels are coming from the structure of the model, or the structure of how it is solved.

**Questions:**

How would a given learning rate profiles influence your conclusions? I think a nonlinear time variable would do the trick. Can it change the qualitative behaviour as well, as the time dependent NDK?

The large N limit, with data size fixed to finite value leads automatically to overfitting, as the weights of the connection to the a neutron are a template of the input. With random weights of infinite extent, there will always be a fitting template for a particular data item. Generalisation suffers, as it would require an efficient compression that finds common templates for multiple data items. In the infinite N limit, this does not happen. Is there a way to look at the large N limit in such a way that one also includes the interesting large data limit?

---

> ### Author Response · Authors · 2023-11-20
>
> We thank the reviewer for raising interesting questions regarding the learning dynamics. We would like to make several clarifications that may potentially address the reviewer's concern. We would like to point out that that our theory considers a small fixed constant learning rate throughout the dynamics (in the notation of the paper, the learning rate is proportional to $1/\lambda$). The two learning stages emerge not because of reduction in the learning rate but due to the difference in the magnitude of the gradient at different points in the loss landscape. Initially when loss is large the gradient term in the Langevin dynamics dominates. When the system reaches the neighborhood of the energy minimum ($\mathcal{O}(T)$ loss), the residual gradient is small, $\mathcal{O}(\sqrt{T})$, of the same size as the Langevin noise, hence the network executes a slow and noise-driven exploration of the solution space, corresponding to a roughly flat region of the loss landscape. Therefore the two phase phenomenon is not a result of adapting learning rate. It reflects the process of an over-parameterized network reaching and exploring a degenerate space of solutions.
>
> Regarding the discretization error, the theoretical results in the paper including the two phases address the continous time stochastic ODE of Langevin dynamics hence does not incorporate discretization errors. It is true that for comparing the theory with simulations of these ODEs we used discretised time Langevin dynamics which suffers from discretization errors. We have fixed the discretization step size so that further reductions of its value did not change significantly the results. Indeed, despite the inevitable discretization step the numerics agree remarkably well with the theory as shown in Fig.1e and SI Fig.7. Relevant work on discretization error of Langevin dynamics suggests that the Wasserstein distances between the original Langevin diffusion and its discretized version decays as $\mathcal{O}(\eta)$ with $\eta$ denoting the discretization step size under appropriate smoothness conditions (Alfonsi et al, 2015); we include the relevant reference in the revised paper.
>
> Response to questions:
>
> 1. Incorporating time dependent learning rate, as suggested by the reviewer, is in principle possible and is an interesting topic for future work.
>
> 2. Various previous works have observed that despite operating in a highly over-parameterized regime, learning tend to bias neural networks towards solutions that also generalize well. This type of 'inductive bias' comes in various forms including explicit and implicit regularizations (Bietti et al, 2019). In our paper, we mitigate the problem of overfitting by adding an explicit $L_{2}$ regularizer, as given in Eq.3 and noted below it. In the long time limit, the network explores the degenerate solution space with small ($\mathcal{O}(T)$) training error, biased towards weights with smaller $L_{2}$ norm, and thus reduces the overfitting problem, as has been shown in standard works on NTK and NNGP in the infinite width and finite data size limit. However, we agree with the reviewer that it would be very interesting to extend the current work to incorporate the more realistic 'linear' regime (where data size scales linearly with width). This is a topic of ongoing work, and is discussed in the discussion section.

---

> > ### Comment · Reviewer_Bpa7 · 2023-11-20
> >
> > I thank the authors for good and thorough clarifications to the reviewers comments. I retain my opinion that this is a good manuscript that deservers to be published in ICLR.
> >
> > I have a question on the development of the gradient when the system closes the minimum.  The behaviour of the gradient in closing in is often reflecting a saddle point type of manifold, where there may be steep slopes in the loss function that will not necessarily depend on how close you are off the minimum.  To remedy this, one is often using caps on the size of the gradient to keep the SGD loss smooth.  Of course, the flatness of the gradients would provide robustness for the inference, so there are benefits in trying to achieve it. In fact, this can be achieve by introducing Hessian dependence into the training loss.
> >
> >  Why would you expect a smooth gradient close to the minimum by default?

---

> ### Author Response · Authors · 2023-11-23
>
> We would like to thank the reviewer for the additional comment. It is true that in general there may be multiple local minima and steep slopes in the loss landscape. In our paper we focus on the infinite width limit, and previous works [1,2,3] have shown that the loss landscape of infinitely wide neural networks has nice properties which ensure smoothness around the minimum and guarantee the convergence of (S)GD algorithms with a fixed learning rate (i.e., without having to cap the gradient, etc).
> Furthermore, these works have shown that for over-parameterized networks, the solution space forms a manifold instead of distinct local minima, which allows us to consider our diffusive learning phase as explorations within the solution manifold.
>
> [1] On the linearity of large non-linear models: when and why the tangent kernel is constant, Liu et al, NeurIPS 2020.
>
> [2] Loss landscapes and optimization in over-parameterized non-linear systems and neural networks, Applied and Computational Harmonic Analysis, Liu et al, 2022.
>
> [3] The loss landscape of over-parameterized neural networks, Cooper, 2018.

---

### Official Review · Reviewer_2yMn · 2023-10-27

**Soundness:** 3 good
**Presentation:** 1 poor
**Contribution:** 4 excellent
**Rating:** 3
**Confidence:** 3

**Summary:**

The authors propose a generalized framework, that includes the Neural Tangent Kernel and the NNGP as special cases. This is facilitated by adding a noise term to the training dynamics (Langevin equations), and then expressing the predictor moments in the infinite-width limit as path integrals over the Langevin trajetories. Theory and experiments investigate training dynamics, finding a gradient-driven phase (NTK) and a diffusion-driven phase (NNGP in long-term equilibrium) in time. Generalization experiments in a classification task then show how noise-related hyperparameters related to initialization and regularization allow to trade the time scales of the two distinct phases against each other. The role of depth and non-linearity is briefly discussed. Lastly, the findings are related to a hypothesis of read-out weight realignment, which was proposed for explaining representational drift observed in neuroscience. For this, the read-out weights are frozen after sufficient training, while the hidden weights are allowed to further diffuse. Experiments show that, depending on whether the kernel is meaningful for the task, the outputs of training and test data decorrelate over diffusion time.

**Strengths:**

(1) A generalized framework is proposed, incorporating both NTK and NNGP. This is highly relevant, as both frameworks have led to major insights into NN theory. Such a generalized framework is of utmost importance and may lead to further novel deep insights.
(2) The approach through treating the necessary noise-injection via Langevin dynamics and path integral formulations is novel and innovative, and may pave the way for a new tool to investigate learning systems via kernel representations. The trick with the markov proximal learning together with the replica trick may even be applied to much more general learning systems.
(3) Interesting original experiments are shown

**Weaknesses:**

(1) Clarity / presentation seems a bit unfocused or maybe even unripe. E.g. the paper's main theoretical result is Eq. 5, only which then allows to elucidate the relations to NNGP and NTK. This Eq. 5 is formulated in terms of random variables \vec{v} and \vec{u}. While \vec{v} is briefly described in prosaic manner, \vec{u} is not explained at all. It is burried deep in the supplementary. Another example is the statement in some regime "the integral equation can be transformed into a linear ODE". This should be supported/elaborated, and if done so in the supplementary, it needs a precise pointer where at exactly. Another example is that the Appendix A is never mentioned anywhere, albeit a major aspect of the theory outlined in Appendix B, as otherwise the path integrals cannot be easily factorized. The paper's originality and meaning is difficult to understand without studying the supplementary in detail.

**Questions:**

(1) Figure 3 is not clear, can you kindly add some more explanations esp. wrt. early stopping and what you mean with it precisely
(2) In the main body (introduction), it is stated that the infinite-width limit is pursued but without requiring the amount of data to scale as the width of the NN. I.e. with finite data. This is the typical NTK or NNGP regime. However, for the main result eq. 5 the replica method is used. To my understanding, the replica method typically relies on a scaling of the data with the NN width. However, the necessity of an infinite-data simultaneous limit is never discussed in the main paper, but is also clearly departing from NTK and NNGP. This seems to indicate a contradiction or at least creates confusion. Please clarify.
(3) In SI eq. 27, the exchange of integral and product of d\theta_\tau is done one line too early.

---

> ### Author Response · Authors · 2023-11-20
>
> We are grateful for the reviewer's comments highlighting the unclear sections of our original paper. We agree that the presentation of Eq. 5 in the previous version of the paper may lead to confusion. In our revised version, we present the moment-generating function for the predictor statistics in Eq.5 and Eq.6, and clarify that ${\bf u}(t)$ and ${\bf v}(t)$ are Gaussian auxiliary integration variables that are integrated out. We have also clarified potential ambiguities in the notations of kernels (see response to Reviewer 2CVF). We have also clarfied the statement "the integral equation can be transformed into a linear ODE" in Section 3.1.We revised the manuscript to incorporate references to SI A, which introduces the Markov proximal learning framework to formulate our theory of Langevin dynamics, and the details of the theoretical calculations is in SI B1-B4. With these changes we believe the statements and derivations of our results are clarified.
>
> Response to questions:
>
> 1. We define optimal early stopping time by the time the network reaches the minimal generalization error in the entire learning trajectory. In Fig. 3, we plot $\Delta t$, defined as the difference between optimal early stopping time and the time when the network first reaches the NNGP equilibrium. We have included a more careful definition in Section 4.2 “The role of initialization and regularization and early stopping phenomena”, and the caption of Figure 3.
>
> 2. Replica: The reviewer is right that in most applications of replica method in statistical mechanics of learning and memory, it is required in the linear regime where the data size and network width are proportional to each other. Relatedly, the use of replica in these works is used to average over the data statistics (assumed to obey simple statistics, e.g., iid gaussian). Our use of the replica method is very different. We do not make assumptions about the statistics of the data at all, hence replica method is not required for dealing with the data. In contrast, we apply the replica method to handle the normalization factor $Z(\theta_{t-1})$ in the transition matrix $\mathcal{T}(\theta_{t}|\theta_{t-1})$ (see details in SI B1), analogous to the application of the replica method in calculation of the Franz-Parisi potential (Franz et al, 1992). This is needed even in the infinite width, finite data limit, in our Markov chain formalism. As the reviewer commented, for this paper we focus on the regime where the network width N tends to infinity, while the amount of data remains finite, consistent with the typical NTK/NNGP regime.
>
> 3. In our revised paper, we add some further clarification for the replica method in SI Eq. 28.

---

> > ### Comment · Reviewer_2yMn · 2023-11-21
> >
> > I thank the authors for their clarifications. I still believe this could be high potential work. However, in its current presentation this paper's impact falls back much below its potential (still). For this reason, I retain my score.
> >
> > It is duly noted that the authors addressed specific points. This reviewer certainly failed in giving a more exhaustive list of examples from the beginning. This was also not the intention; rather that the presentation would benefit from a larger revision. I will give further examples around the main theoretical result to illustrate my point:
> >
> > - In the revised version, the main theoretical result eq. 5+6 introduces a "field coupled to the predictor \ell(t)". No notion is given what this means, what this field is, what the coupling looks like, or why it is important / necessary. It then appears again later somewhere in the body of the supplementary as \ell_t around eq. 30 (eq 27 in old version), which is the first point where one gets an idea of what this object is
> > - These equations have actually changed from the last version. Why? (\tilde u instead of u)
> > - In these equations, an object m(t,t') is used to define the action. Only a couple of equations and one section later, m(t,t') is actually introduced, and only in the context of having introduced the kernel already. While this is clearly simple to fix, it exemplifies the early stage of this manuscript's presentation
> > - A minor effort has been made to clarify the introduction of the auxiliaries u(t) and v(t), but still not clear

---

> > > ### Author Response · Authors · 2023-11-23
> > >
> > > We appreciate the reviewer for pointing out the weakness with our presentation of results.
> > > - In the revised manuscript, the main theoretical result is a moment generating function, and one can evaluate moments of the predictor by taking derivative of the MGF w.r.t. $\ell(t)$, at $\ell(t)=0$. We elaborated this in our current version.
> > > - In the previous version, the equation was not the moment-generating function, instead, it is $S({\bf u}(t), {\bf v}(t))$  in the exponent of the moment generating function (MGF). We have modified it as we believe introducing the MGF is more clearly motivated, as we use the MGF to derive the mean predictor. $\tilde{\bf u}(t)$ is introduced as stated before Eqs. 5 and 6, which is a $P+1$ dimensional vector, given by ${\bf u}(t)$ concatenated with $\ell(t)$.
> > > - In our paper, we introduce the MGF first, and clarify that this can be used to derive the mean predictor. It is difficult to define all terms in the MGF immediately after it. Therefore, we add a brief paragraph introducing each components in the MGF, and devote the next section to introduce the detailed definition of each term, including $m(t,t')$ and the kernel functions.

---

### Official Review · Reviewer_88ey · 2023-10-31

**Soundness:** 3 good
**Presentation:** 4 excellent
**Contribution:** 4 excellent
**Rating:** 8
**Confidence:** 3

**Summary:**

The authors study the learning dynamics of (regularized) infinite-width neural networks under Langevin dynamics (continuous stochastic gradient descent). They derive a new kernel termed the neural dynamic kernel (NDK) which exhibits both NTK and NNGP behavior in the limits. NDK can be thought of as the two-time extension of NTK under Langevin dynamics. Consistent with empirical results it explains two phases of dynamics, a deterministic gradient-driven phase (akin to NTK) followed by a diffusive phase exploring the solution space and approaching the posterior distribution of an infinite-width NN with Normal prior over the weights (akin to NNGP). Empirical results are in line with theoretical arguments and explore the two phases of learning dynamics as well as the effect of depth, temperature, initialization variance, and equilibrium variance. Furthermore, the authors use the theoretical framework and make connections to the representational drift phenomenon observed in neural systems which corresponds to the reorganization of synaptic weights while maintaining the behavioral performance in experimental tasks.

**Strengths:**

There has been a large interest in the behavior of overparameterized neural networks in the ML community. NTK and NNGP were introduced as two seemingly separate tools for studying the limiting behavior of infinite neural networks in the deterministic and Bayesian settings. While previous literature has pointed to connections between the two frameworks a clear unified framework for understanding the two didn't exist. This paper tackles this problem by introducing NDK and studying the limiting behavior.

The empirical results make the theoretical arguments very clear. Depicting the two phases of dynamics along with the limiting behavior of the models provides compelling evidence for the theoretical framework.

The connections to the neuroscience literature on representational drift are insightful and interesting (although this paper is not the first to make this connection and other papers in the literature on overparameterized neural nets already made this connection before) [1,2].

[1] https://arxiv.org/abs/2110.06914

[2] https://arxiv.org/abs/1904.09080

**Weaknesses:**

My main comment is about the introduction part of the paper which can be made more comprehensive. The literature on learning dynamics in overparameterized neural nets has progressed quite extensively and learning dynamics, and placing the work in the existing literature and recognizing existing contributions fairly will make the paper stronger.

Although I think the two-time extension of NTK is a novel contribution, I’ve seen the one-time extension of NTK before (see e.g. Lemma 8.1.1 of [1]). In fact, the development of the NTK is the consequence of the limiting behavior of this one-time extension.

The authors cite a paper that discusses the diffusive properties of Langevin dynamics in the long-time limit. Another paper that discusses this diffusive behavior under learning using SGD is [2]. It appears to me that this is not a novel finding as it was reported before and it's a phenomenon that's already justified using theoretical arguments. Although this paper provides a unique perspective connecting NTK and NNGP using NDK I would frame the diffusive behavior of the SGD as a confirmation of the existing literature rather than a novel finding.

[1] https://www.cs.princeton.edu/courses/archive/fall19/cos597B/lecnotes/bookdraft.pdf

[2] https://arxiv.org/abs/2110.06914

**Questions:**

- Can the authors discuss the extensions to multivariate outputs?
- Can the framework be extended to different forms of regularization? Specifically, do we expect to observe the diffusive behavior and the limiting posterior distribution of a Bayesian model with a different prior under a different regularization (e.g. l1 regularization)?
- The authors argue the relevance of SGD in the context of batch learning, does this framework have the potential to explain other tricks used in the learning of neural nets such as dropout, batch-norm, etc.?
- Can we derive bounds w.r.t. the number of iterations, depth, number of training samples, etc. on when to expect to observe the transition from phase 1 to phase 2 and use that as the early stopping criterion?

---

> ### Author Response · Authors · 2023-11-20
>
> We appreciate the reviewer for pointing out the several interesting and relevant works, we have modified the introduction part of the paper to recognize and connect to existing contributions, and included relevant references as the reviewer suggests. With respect to previous single-time NTK kernel (now cited in the Introduction) we note that the time dependence of NTK is not explicit but arises through a weight dependent kernel (i.e., before any averaging). In contrast, our NDK has an explicit time dependence that arises when the weights are averaged. As we show, the full dynamics requires averaging over two sets of weights (i.e., at two time points) hence two times.
>
> Response to questions:
>
> 1. Multiple outputs: Thanks for asking. As we comment now in discussion section of the paper, for $M$ outputs where $M\sim\mathcal{O}(1)$, one can simply replace the $P$ dimensional vector ${\bf y}$ by a $P\times M$ dimensional matrix in Eq.14 and 15.
>
> 2&3. We thank the reviewer for these interesting questions regarding extensions of our current results. $L_{1}$ regularization is challenging as the integration over the weight priors is harder. Effects of batch norm and dropout can in principle be incorporated by adding state dependent noise (Yang, 2019; Schoenholz et al.,2017). These interesting extensions are topics for future work.
>
> 4. The transition from phase 1 to phase 2 happens after the network reaches the NTK equilibrium. As Eq.16 (revised) and previous work (Jacot et al, 2018) show, the convergence time of the NTK dynamics depends on the spectrum of the NTK, which is determined by the data structure, number of training examples, network nonlinearities, and network depth. This allows us to evaluate where the transition happens. However, the early stopping time, defined as the time where the network reaches minimum generalization error across the entire learning trajectory is more complicated. As we show in Fig.2c and Fig.3, it may occur after the transition to phase 2, and we do not have a simple way of determining the early stopping point without numerically solving Eq.14 and 15.

---

> ### Comment · Reviewer_88ey · 2023-11-21
>
> I thank the authors for further clarifications. I just noticed that in the revised manuscript under the contributions sections, there are two no 4's, please fix that. Also, I'd like to stress a previous comment that I made, regarding the connection between the diffusive properties of Langevin dynamics and representational drift that was discovered in previous work (check the two papers I cited under "Strengths". If you agree with this, please fix the point 4 contribution and rewrite it as reaffirming the existing connection as opposed to being the first ones to make this connection.
>
> I maintain my score and I believe the work deserves to be published.

---

> ### Author Response · Authors · 2023-11-23
>
> We thank the reviewer for stressing this point. We have made the changes, and added a point 5 contribution to discuss relation with previous works. We added reference to the relevant literature, and phrased it as "Our work complements these previous findings by providing theoretical analysis of the two phases under Langevin dynamics, reaffirming the connections between the diffusive learning stage and representational drift in neuroscience as established in previous works".

---

### Official Review · Reviewer_2CVF · 2023-11-01

**Soundness:** 3 good
**Presentation:** 2 fair
**Contribution:** 3 good
**Rating:** 5
**Confidence:** 5

**Summary:**

This paper explores the connection between deep neural networks and kernel learning. Specifically, the authors take a representation of the dynamics of evolution of the hyper-parameters in what is presumably a neural network and study the evolution of the kernel corresponding to the feature space. The authors show that the resulting kernel evolves from the NTK to the NNGP kernel (or at least that is my interpretation).

**Strengths:**

While there is obviously a significant body of work on convergence to the NTK, the connection from NTK to NNGP under stochastic noise is an interesting hypothesis. The authors explore the implications from several perspectives -- including testing the proposed NDK for different regimes and some interesting outcomes such as early-stopping and representation drift.

**Weaknesses:**

My major concerns with the paper, however, are not on the idea itself, however, but rather that the fundamental results are not rigorously established before proceeding to interpretation. This makes it hard to quantify or verify the claims. Specifically, the paper has no proofs or definitions which can be verified. There are many claims being made and several equations used to back up those claims. However, without rigorous definitions and statements to prove, it is difficult to verify these claims. In addition, the notation is inconsistent and meaning is unclear at times.

My rating for this paper is based on the anticipated magnitude of required restructuring and revision.

**Questions:**

Some detailed notes and questions supporting the summary are as given below.

1) The authors make constant use of the first person plural in a way which does not encompass the reader. This makes it seem as if the paper were about the authors and not about the result.

2) Section 2.2 has a lack of definitions. For example, the variable $\mathbf{u}(t)$ --  what are the particular properties of $u$ for which  Equation $5$ holds?

3) Some motivation for NDK kernel is presented in Equation 5, but Equation 5 has not been properly introduced.  Authors should include at least basic information describing equation 5 in Section 2.2 and show why this statistic $S$ is important.

4) There is quite a bit of inconsistent notation. For example, Equation 6 has a function $\nabla_\theta f(x, t)$ and then Equation 9 uses n $f(x, \theta)$. In another example, $K$ has variously four arguments and then three and then two.

5) The authors often use $\dot K(t,t',x,x')$. But what is the derivative with respect to?

6) Equation 16 should be described in more detail. If Equation 16 is just a classical NNGP result, then the connection between NDK and NNGP is unclear. If Equation 16 is the limit of Equation 13, then this result should be presented in the paper as well, since Equation 13 depends on $\mathbf{f}_{\text{train}}(t')$.

7) If we consider $t = t'$ for NDK, how is the dynamic different from NTK?

8) In Fig 2, it would be nice to add NNGP equilibrium for comparison.

9) How do authors define the optimal early stopping point and  optimum generalization error?

10) The font of labels, legends and titles of all figures should be increased.

11) ``All the kernel functions above including the two-time NNGP kernel, the
derivative kernel, and the NDK, have closed-form expressions for specific activation functions such
as linear, ReLU and error function (see SI Sec.C, Cho & Saul (2009)).'' -- which equations are the authors referring to?

12) Supplementary information such as Sec. B provides little to no explanation or context.

13) ``predictor converges fast to the desired''

14) ``Our theoretical framework for the Langevin dynamics provides a model of representational drift'' Which framework? The model? This is unclear.

15) Margin violation on Eq 88

---

> ### Author Response · Authors · 2023-11-20
>
> We thank the reviewer for questions regarding the clarity of our paper. We have modified the paper to clarify the notations, equations and references to the appendix.
>
> Response to questions:
> 1. We acknowledge your concern. In our revision, we will carefully address this issue and make necessary adjustments to reduce the use of first person plural and emphasize the significance of the findings.
>
> 2 and 3.  We appreciate the feedback and we agree that Eq. 5 may lead to confusion in its previous version. In our revised version, we present the moment-generating function (MGF) for the statistics of the predictor $f({\bf x},t)$ in Eq.5 and Eq.6, and ${\bf u}(t)$ and ${\bf v}(t)$ are auxiliary integration variables which are integrated out. Eq. 5 is important because the predictor statistics, and in particular the mean predictor can be derived from it by evaluating its derivative w.r.t. the field $\ell(t)$ at $\ell(t)=0$. We have modified the paper and incorporate necessary explanations of Eq.5 to highlight its importance and its connection to the kernel functions and predictor statistics.
>
> 4. The typo in Eq. 9 of the original paper has been fixed, it is now Eq.10 in the revised paper. We apologize for the confusion of multiple notations for the kernel function applied on training/testing data. In the revised version, we clarify the kernel notations by using bold capital letters ${\bf K}(t,t')$ to denote  $P\times P$  kernel matrices, ${\bf k}(t,t')$  to denote $P$ dimensional kernel vector, and $K(t,t',\boldsymbol{x},\boldsymbol{x}')$ to denote functions of two arbitrary inputs ${\bf x}$ and ${\bf x}'$. We also denote by ${\bf \tilde{K}}(t,t')$ $(P+1)\times(P+1)$ or $(P+1)\times P$ matrices (where the $P+1$ index refers to the test point). These notations are detailed at the end of Section 2.2 and Section 2.4.
>
> 5. The derivative kernel is defined as Eq.13 in our revised paper.
>
> 6. Eq. 17 is the limit of Eq.12 which relates the NDK to the NNGP kernel. It results from the definition of the NDK and not from the equation for the predictor. This non-trivial identity is proven in SI Sec. C4.
>
> 7. NDK with $t=t'$ has the form of a time-dependent averaged NTK(Eq.7). In the infinite width limit, the NTK theory states that the learning dynamics is govened by the NTK at initiliaztation (Jacot et al, 2018), hence does not have time dependence. Indeed, as we show in Eq.16, in the relevant time scale appropriate for the NTK theory, NDK with $t=t'$ is independent of time. However, for longer time scales NDK with $t=t'$ is not equivalent to NTK (Eq.12), specifically, our NDK is explicitly time dependent since we averaged over the weights $\Theta(t)$, while the NTK implicitly depends on time through $\Theta(t)$. Also, as shown in Eq. 15, the predictor dynamics depend on the NDK with general $t$ and $t'$.
>
> 8. We already show the NNGP equilibrium in Fig. 2, as shown by the red dashed lines.
>
> 9. We appreciate the reviewer for pointing out the unclear aspects of the definition of the optimal early stopping point, we have correspondingly modified the relevant paragraph in section 4.2 for clarity. The optimal early stopping point for a set of test data points, is defined as the time where generalization error on this set achieves its minimal value (as a function of time).
>
> 10. We will increase the font of labels, legends and titles of figures in our finalized version.
>
> 11. For linear activation the expressions for the kernels are straightforward as shown in SI Eq. 78-80. For ReLU activation, we are refering to Eq. 12 and 13 in Cho and Saul 2009, with $n=1$, derived in SI A of the paper, and also shown in SI Eq. 81-85 in our paper. For error function, we refer to Eq. 11 in Williams 1996, and also shown in SI Eq. 86-87 in our paper.
>
> 12. We have included a paragraph at the beginning of SI Sec. B outlining the content of this section.
>
> 13. It is unclear what the reviewer is asking, but we have changed the sentence to “predictor converges fast to the target output” for clarification.
>
> 14. We agree that this sentence is confusing. We have changed it to “our theory for the Langevin dynamics suggests a possible mechanism of representational drift”.
>
> 15. We thank the reviewer for noticing this, we have fixed the margin violation problem.

---

> ### Comment · Reviewer_2CVF · 2023-11-20
>
> I have reviewed the author response. However, I do not believe the anticipated revisions will be enough to satisfy the substance of my concerns.

---

### Meta-Review · Area_Chair_udw7 · 2023-12-05

**Metareview:**

This paper considers the learning dynamics of infinite-width neural networks under continuous time stochastic gradient descent. The authors derive a novel kernel, which can be seen as a time-dependent version of the NTK, and includes both the NTK and NNGP as special cases. All the reviewers agreed that the contribution brings an innovative and useful perspective on both the NTK and NNGP. There were diverging views amongst reviewers regarding whether or not the paper should be accepted. The reviewers noted that the presentation of the paper was overall weak, suffering from missing notations, typos, etc. While the authors have corrected some of them in the revised version, it was felt by two reviewers that the paper would benefit from further polishing. I agree with their assessment, and recommend a rejection.

**Justification For Why Not Higher Score:**

A large number of typos and notational issues were noted by two reviewers. The paper would require a bit more polishing.

**Justification For Why Not Lower Score:**

N/A

---

### Decision · Program_Chairs · 2024-01-16

Reject